# Dopamine receptors reveal an essential role of IFT-B, KIF17, and Rab23 in delivering specific receptors to primary cilia

**Alison Leaf[1,2], Mark Von Zastrow[1,3,4]***

[1]Program in Cell Biology, University of California, San Francisco, San Francisco, United States; [2]Department of Biochemistry and Biophysics, University of California, San Francisco, San Francisco, United States; [3]Department of Psychiatry, University of California, San Francisco, San Francisco, United States; [4]Department of Cellular and Molecular Pharmacology, University of California, San Francisco, San Francisco, United States

**Abstract** Appropriate physiological signaling by primary cilia depends on the specific targeting of particular receptors to the ciliary membrane, but how this occurs remains poorly understood. In this study, we show that D1-type dopaminergic receptors are delivered to cilia from the extra-ciliary plasma membrane by a mechanism requiring the receptor cytoplasmic tail, the intraflagellar transport complex-B (IFT-B), and ciliary kinesin KIF17. This targeting mechanism critically depends on Rab23, a small guanine nucleotide binding protein that has important effects on physiological signaling from cilia but was not known previously to be essential for ciliary delivery of any cargo. Depleting Rab23 prevents dopamine receptors from accessing the ciliary membrane. Conversely, fusion of Rab23 to a non-ciliary receptor is sufficient to drive robust, nucleotide-dependent mis-localization to the ciliary membrane. Dopamine receptors thus reveal a previously unrecognized mechanism of ciliary receptor targeting and functional role of Rab23 in promoting this process.

*For correspondence: Mark. VonZastrow@ucsf.edu

**Competing interests:** The authors declare that no competing interests exist.

## Introduction

Primary cilia are microtubule-based protrusions of the plasma membrane that support a wide range of specialized receptor-mediated signaling functions. Physiological signaling from cilia critically depends on the selectivity of receptor targeting to the ciliary membrane, and disturbances in this targeting are thought to underlie a variety of pathological conditions (*Hsiao et al., 2012*). The remarkable specificity of ciliary membrane targeting is clear among G protein-coupled receptors (GPCRs). Some members of this large receptor family robustly accumulate in the ciliary membrane while others, including closely related homologues, are found throughout the extra-ciliary plasma membrane but are effectively excluded from cilia (*Schulz et al., 2000*; *Marley and von Zastrow, 2010*). Understanding how particular GPCRs localize to primary cilia with such exquisite selectivity is a fundamental problem with broad physiological significance (*Emmer et al., 2010*).

The ciliary membrane compartment is separated from the surrounding extra-ciliary plasma membrane by a transition zone complex that impedes lateral exchange of membrane proteins (*Gilula and Satir, 1972*; *Hu et al., 2010*; *Chih et al., 2011*; *Williams et al., 2011*). This can explain how GPCRs are retained in cilia once delivered, but not how they are delivered in the first place. Two

**eLife digest** Slender structures called primary cilia protrude from the outer membrane of nearly every human cell. Each cell generally has one cilium, which helps the cell to sense its environment and respond to the signaling molecules sent to the cell to influence its behavior. Proteins called receptors, which are embedded in the surface of the cilia, bind to these molecules and help to transmit information about these signals into the cell.

The receptor proteins are not made in the cilia. So far, two broad ways of transporting the receptors to the cilia have been established: some receptors are carried through the interior of the cell inside small membrane-enclosed compartments, and some are pulled from another part of the cell's membrane. However, the exact steps and molecules involved in transporting different receptor types are not fully known.

One type of receptor that is commonly found in the cilia of many different cells interacts with a signaling molecule called dopamine. Leaf and Von Zastrow tagged the dopamine receptor with a fluorescent protein that allowed its movement through a cell to be followed under a microscope. This revealed that the dopamine receptor slides into cilia from another part of the cell's membrane. To do so, the dopamine receptor binds to a pair of proteins known to help move receptors around the cell. A third protein called Rab23 then acts as a gatekeeper that allows the dopamine receptor to pass into the cilia. Reducing the amount of Rab23 in a cell prevented dopamine receptors from entering the cilia. Furthermore, Leaf and Von Zastrow found that other receptor proteins incorrectly moved into the cilia if they were bound to Rab23.

Rab23 was already known to play a role in controlling signaling in cilia, and mutations in the Rab23 gene cause disease in humans, but Rab23 was not known to be necessary for delivering receptors to cilia. Future studies to further reveal how Rab23 directs receptors to cilia, and to determine which other ciliary proteins might use this transportation process, will lead to a better understanding of the normal structure and activity of primary cilia. This could lead to new strategies for treating the human diseases that involve defective cilia.

basic routes of ciliary membrane delivery have been described: first, receptors can originate from an intracellular source, through fusion of post-Golgi transport vesicles with the ciliary membrane in or near the transition zone. A number of membrane proteins are targeted to cilia by this route, and molecular machineries supporting it have been identified (*Deretic and Papermaster, 1991*; *Geng et al., 2006*; *Mazelova et al., 2009*). Second, receptors can originate from the extra-ciliary plasma membrane. This route, first described in a seminal study of flagellar agglutinins in *Chalmydomonas* (*Hunnicutt et al., 1990*), contributes to ciliary targeting of the atypical seven-transmembrane protein Smoothened (Smo; *Milenkovic et al., 2009*) in mammalian cells. Is the lateral delivery route relevant to ciliary localization of conventional GPCRs?

Molecular mechanisms that underlie specific ciliary delivery pathways also remain incompletely understood. A number of proteins are already known to play a role, including the BBSome (*Nachury et al., 2007*; *Berbari et al., 2008b*; *Jin et al., 2010*), Tulp3 (*Mukhopadhyay et al., 2010*, *2013*), Arf4 (*Deretic et al., 2005*), ASAP1 (*Wang et al., 2012*), and intraflagellar transport (IFT)-B and IFT-A (*Mukhopadhyay et al., 2010*; *Keady et al., 2011*, *2012*; *Crouse et al., 2014*; *Kuzhandaivel et al., 2014*). Are there additional machineries not yet identified that function in targeting specific GPCRs to cilia?

We addressed these questions through study of the D1-type dopamine receptor (D1R), a conventional GPCR that robustly localizes to cilia in diverse cell types (*Marley and von Zastrow, 2010*; *Domire et al., 2011*). Here, we show that D1Rs are delivered to the cilium from the extra-ciliary plasma membrane. Further, we show that the D1R cytoplasmic tail is both necessary and sufficient to direct receptor targeting to the ciliary membrane, and this requires a distinct set of cellular proteins including the anterograde IFT-B complex and ciliary kinesin, KIF17. Moreover, we identify an essential role of the small GTP-binding protein, Rab23, in the ciliary targeting mechanism. Rab23 is not only necessary for D1R access to cilia, it is also sufficient to drive strong ciliary localization of a non-ciliary GPCR. D1Rs thus reveal a discrete route and mechanism of ciliary GPCR targeting in which Rab23 plays an unprecedented and essential role.

## Results

### D1Rs are robustly targeted to the primary cilium

The D1R is a cilia-localized GPCR whose mechanism of targeting to the cilium is poorly understood (*Marley and von Zastrow, 2010*; *Domire et al., 2011*; *Zhang et al., 2013*). We investigated this question using recombinant receptors expressed in inner medullary collecting duct (IMCD3) cells. Using an N-terminal Flag tag on the D1R to label the overall surface pool, D1Rs were visualized throughout the plasma membrane and highly enriched in cilia marked by acetylated tubulin (AcTub) (*Figure 1A*), like the cilia-localized somatostatin-3 receptor (SSTR3) (*Figure 1B*; *Händel et al., 1999*; *Schulz et al., 2000*; *Berbari et al., 2008a*). In contrast, the delta opioid peptide receptor (DOP-R or DOR) localized throughout the extra-ciliary plasma membrane but was not detectable on cilia (*Figure 1C*).

We first quantified ciliary localization by counting the number of receptor-expressing cells with visible receptor immunoreactivity on the cilium. This normative metric verified ubiquitous D1R localization to cilia, similar to SSTR3, and high specificity of ciliary localization relative to DOR (*Figure 1D*). Second, because cilia scored as receptor-positive varied in degree of apparent receptor concentration, we determined average fold-enrichment of receptors on the cilium relative to the extra-ciliary plasma membrane (*Figure 1E*). This graded metric further verified robust ciliary localization of the D1R and SSTR3 (but not DOR) and indicated that the D1R is enriched on cilia even more strongly than the SSTR3 (*Figure 1F*).

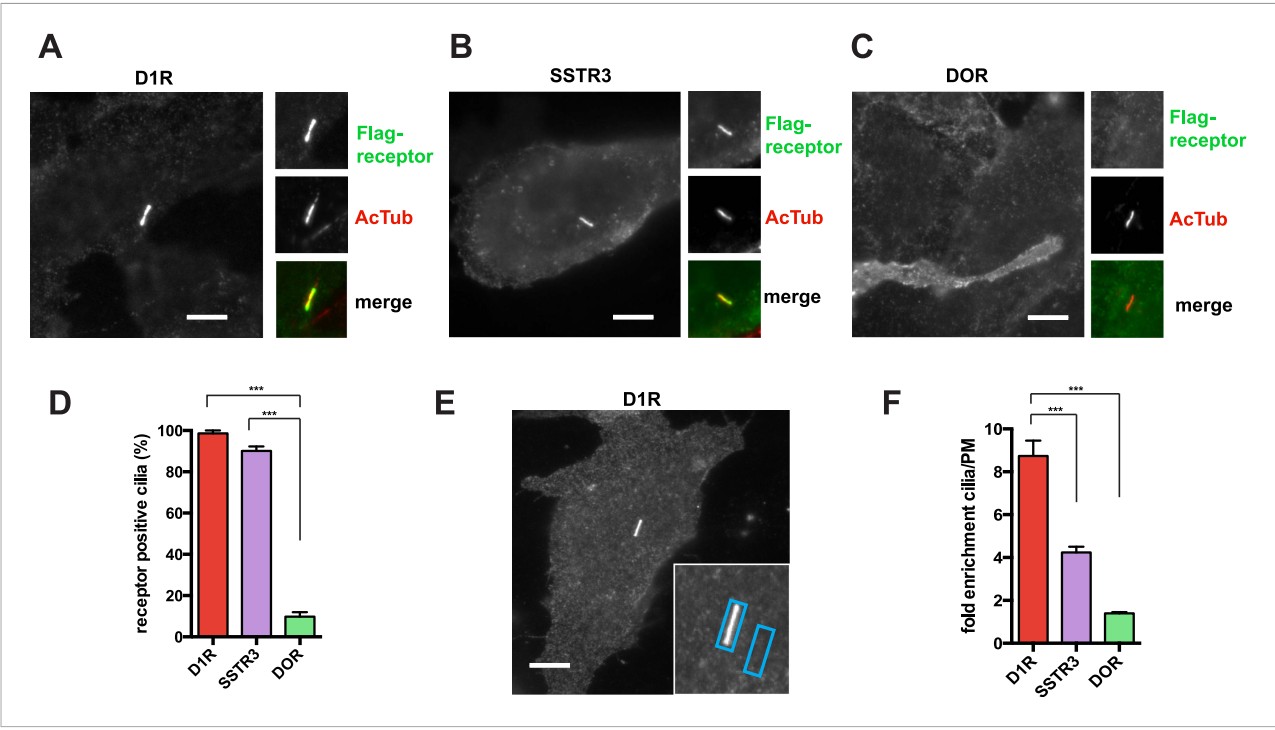

**Figure 1**. D1Rs specifically localize to primary cilia. (**A**–**C**) Representative epifluorescence microscopy images of Flag-D1R (panel **A**), Flag-SSTR3 (panel **B**), and Flag-DOR (panel **C**) localization on the surface of inner medullary collecting duct (IMCD3) cells. Insets show a cropped region of the plasma membrane containing the cilium, with Flag immunoreactivity marking receptor (top) and acetylated tubulin (AcTub) immunoreactivity marking the cilium (middle). Merged view is at bottom with Flag in green and AcTub in red. Flag-D1R and Flag-SSTR3 localize robustly to cilia, while Flag-DOR is detectable in the extra-ciliary plasma membrane but not on cilia. (**D**) Quantification of ciliary localization by determining the fraction of receptor (Flag)-positive cilia, judged by the presence of Flag immunoreactivity visible by epifluorescence microscopy, and expressed as a percentage of total cilia counted in the transfected cell population. (**E**) Scheme for quantification of ciliary localization by determining enrichment of receptor (Flag) signal in an ROI containing the cilium, when compared to an adjacent region of the extra-ciliary plasma membrane. Representative ROIs are shown for a Flag-D1R-transfected cell. (**F**) Fold-enrichment calculated as a ratio of background-subtracted Flag signal present in the ciliary ROI divided by background-subtracted Flag signal present in the adjacent extra-ciliary plasma membrane ROI (cilia/PM). Error bars represent SEM from n = 3 independent experiments, with 10–15 cilia analyzed for each receptor in each experiment. (***) p < 0.001. Scale bars, 5 μm.

## D1Rs are mobile in the ciliary membrane and accumulated by continuous delivery from the extra-ciliary plasma membrane pool

In principle, D1Rs could be concentrated on cilia relative to the extra-ciliary plasma membrane by immobilization or by a diffusion barrier at the base of the cilium (*Huang et al., 2007*; *Hu et al., 2010*; *Francis et al., 2011*). To distinguish these possibilities, we fused the D1R to photoactivatable green fluorescent protein (Flag-D1-PAGFP) and investigated mobility by live cell imaging coupled to local photoactivation (*Figure 2A*). We verified appropriate ciliary localization of engineered receptors by anti-Flag Alexa555 surface labeling (*Figure 2B*, top row). Locally photoactivated D1Rs were distributed non-uniformly on cilia immediately after the 405-nm illumination pulse and then equilibrated throughout the cilium within seconds (*Figure 2B,C*; whole-cell images shown in *Figure 2—figure supplement 1*). However, there was no visible spread of labeled receptors outside of the cilium on this time scale. Consistent with this, total PA-GFP fluorescence intensity integrated over the ciliary length remained unchanged throughout an 80-s interval after local photoactivation (*Figure 2D*). Further, a robust fluorescence signal representing photoactivated D1Rs was still visible when the same cilia were re-imaged minutes thereafter (*Figure 2E*; whole-cell images shown in *Figure 2—figure supplement 2*). Together, these observations indicate that ciliary D1Rs are laterally mobile in the ciliary membrane compartment but restricted from freely diffusing into the extra-ciliary plasma membrane.

Achieving net ciliary concentration of receptors that are laterally mobile requires the ability of new receptors to enter the cilium. We sought to measure this delivery by taking advantage of the irreversible nature of PA-GFP photoactivation (*Lippincott-Schwartz et al., 2003*). Multiple 405-nm light pulses were delivered in rapid succession to photoactivate the majority of D1Rs present in the cilium. We then assessed the effect of administering a subsequent 405-nm pulse at a later time. We reasoned that, because D1Rs delivered from outside the cilium would not have been previously photoactivated, their arrival in the cilium would result in an increment in the ciliary PA-GFP signal elicited by a subsequent 405-nm pulse. This was indeed the case: when a subsequent 405-nm pulse was administered shortly (~30 s) after the initial photoactivation series, little (~15%) increase of ciliary PA-GFP fluorescence was observed, consistent with a small residual fraction of D1Rs escaping photoactivation in the initial pulse series (*Figure 2F*, left bar). However, when the subsequent 405-nm pulse was administered 30 min after the initial series, the increment of ciliary PA-GFP fluorescence was markedly increased (*Figure 2F*, right bar). Additionally, the degree of the PA-GFP fluorescence increment increased with time (*Figure 2—figure supplement 3*), and we verified that the PA-GFP fluorescence increment was negligible when measured in fixed cells (*Figure 2—figure supplement 4*). These results directly verify active delivery of D1Rs to the ciliary membrane compartment and provide a rough estimate of the rate of this delivery, on the order of ~1% of the total ciliary D1R pool per min.

D1Rs delivered to the cilium could originate from an internal vesicular pool or from the extra-ciliary plasma membrane (*Deretic and Papermaster, 1991*; *Milenkovic et al., 2009*; *Wang et al., 2009*). To distinguish these possibilities, we further elaborated the sequential photoactivation technique by taking advantage of dual labeling of surface D1Rs using anti-Flag conjugated to Alexa555, whose fluorescence was not affected by the 405-nm pulses (*Figure 2G*). If D1R delivery originates from an internal membrane pool, we expected (1) unchanged PA-GFP/Alexa555 ratio over the 30-min incubation after initial photoactivation because receptors delivered during this interval would not contribute fluorescence in either channel and (2) increased PA-GFP/Alexa555 ratio above the initial value after the subsequent photoactivation pulse because newly delivered receptors would contribute PA-GFP but not Alexa555 signal. On the other hand, if ciliary D1R delivery originates from a plasma membrane pool, we expected (1) decreased PA-GFP/Alexa555 ratio during the 30-min incubation after initial photoactivation because newly delivered D1Rs would be labeled with Alexa555 but lack PA-GFP signal and (2) return to the initial value after the subsequent 405-nm pulse because newly delivered D1Rs would then contribute both Alexa555 and PA-GFP signal. We observed precisely the latter result: PA-GFP/Alexa555 decreased by approximately 50% during the 30-min incubation after initial photoactivation and returned to a value close to the initial ratio after the subsequent photoactivation pulse (*Figure 2H*; example images from separate fluorescence channels shown in *Figure 2—figure supplement 5*). These observations indicate that D1Rs are delivered to the ciliary membrane compartment primarily from the extra-ciliary plasma membrane pool. Supporting the validity of this fluorescence ratio determination, bleaching of both fluorophores was negligible after sequential image acquisitions exceeding the number required for this experiment (*Figure 2—figure supplement 6*), and

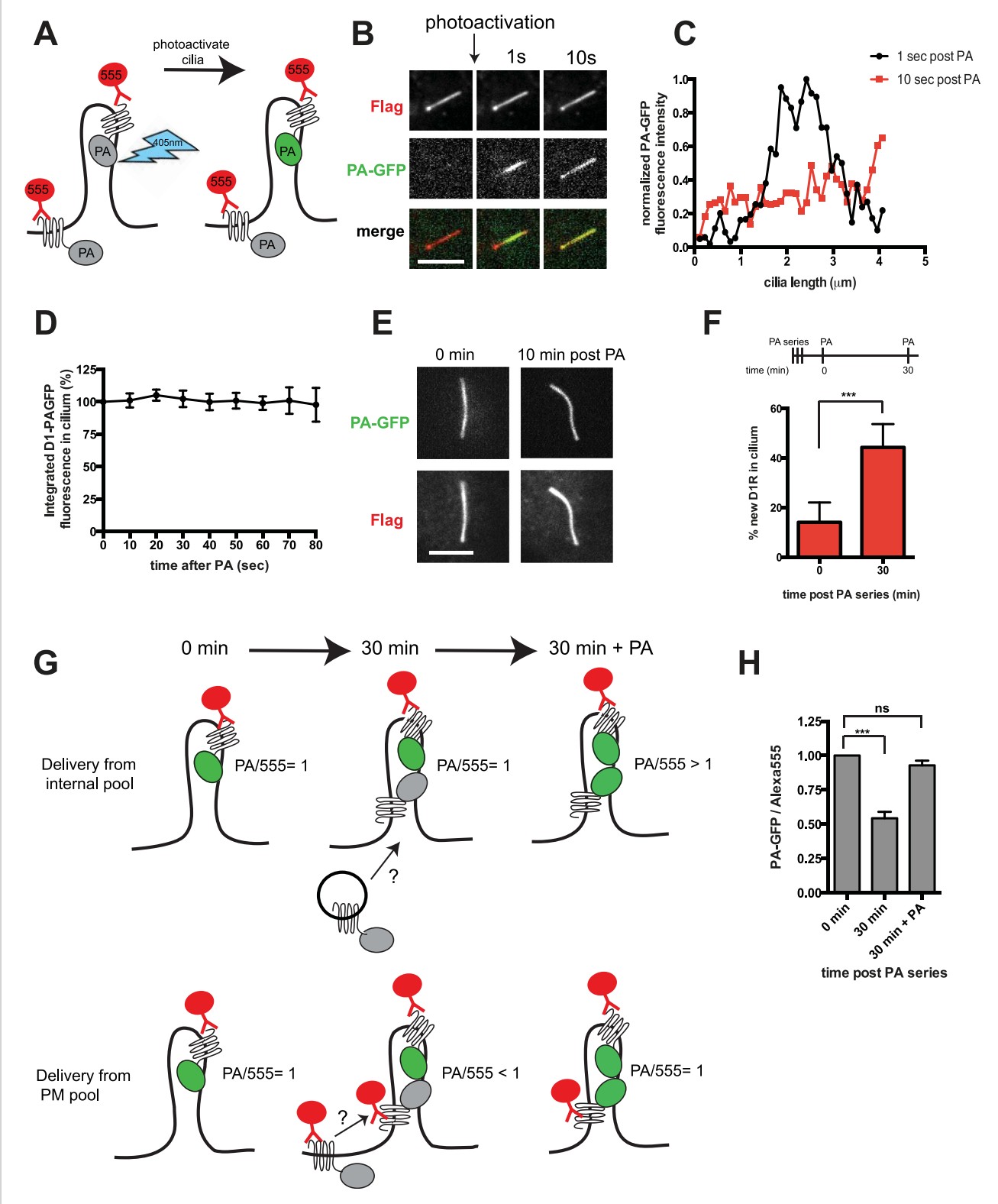

**Figure 2**. D1Rs are mobile in the ciliary membrane and delivered from the extra-ciliary plasma membrane. (**A**) Schematic for local labeling of D1-type dopamine receptor (D1R) in a cilium using PA-GFP. IMCD3 cells expressing Flag-D1-PAGFP were labeled with anti-Flag antibody conjugated to Alexa555 to visualize the overall surface receptor pool. A point-focused 405-nm laser spot was used to locally photoactivate receptors on the mid-portion of the cilium. Non-fluorescent PA-GFP is depicted in gray, fluorescent state in green. (**B**) Live cell confocal images of a representative cilium showing the frame

*Figure 2. Continued*

immediately before the photoactivation pulse (left column), and frames acquired 1 s (middle column) and 10 s (right column) after local photoactivation. The Flag-Alexa555 signal labeling the entire surface receptor pool (top row) was present throughout the cilium at all time points. PA-GFP fluorescence representing the photoactivated pool was non-uniformly distributed at 1 s and uniformly distributed along the cilium within 10 s. (**C**) Line scan analysis of PA-GFP fluorescence along the cilium from the example in panel **B**. (**D**) Integrated PA-GFP fluorescence signal in the cilium as a function of time after the 405-nm laser pulse. The PA-GFP fluorescence at time = 0 was set at 100%. Points represent the mean fraction of PA-GFP fluorescence present in the cilium over an 80-s imaging interval. Error bars represent SD from analysis of n = 6 cilia. There was no detectable loss of ciliary PA-GFP signal quantified over an 80-s interval. (**E**) Confocal images of a representative cilium acquired immediately after (0 min) and 10 min after photoactivation, showing that the locally photoactivated receptor pool was largely retained in the cilium even after this longer interval. (**F**) Assessing new D1R delivery to the cilium by saturation photoactivation and the sequential 'image-photoactivate-image' scheme described in the 'Materials and methods'. Bars represent mean fractional increase in ciliary PA-GFP fluorescence elicited by the subsequent test pulse. Error bars represent SD for n = 7 cilia. (**G**) Schematic for modifying the saturation photoactivation method to assess source of newly delivered D1Rs, based on the ratio of integrated PA-GFP/Alexa555 fluorescence (PA/555) measured in the cilium as a function of time. The initial condition is depicted on the left ('0 min') with the fluorescence ratio (PA/555) arbitrarily set to 1. If new receptors enter the cilium from an internal membrane pool during the 30-min incubation period (depicted in center, '30 min'), they contribute neither Alexa555 nor PA-GFP signal, so the fluorescence ratio is unchanged from the initial condition (PA/555 = 1). After the subsequent 405-nm test pulse (depicted at right, '30 min + PA'), the PA-GFP signal increases without any change in Alexa555 signal, elevating the fluorescence ratio above the initial condition (PA/555 > 1). If new receptors enter the cilium from the extra-ciliary plasma membrane pool, they contribute Alexa555 but not PA-GFP signal during the 30-min incubation, reducing the fluorescence ratio from the initial condition (PA/555 < 1). The subsequent 405-nm pulse restores the fluorescence ratio to the initial value (PA/555 = 1). (**H**) Experimental results from the strategy depicted in panel **G**. Bars represent the mean ratio of integrated PA-GFP/Alexa555 fluorescence measured in the cilium. Error bars represent SD from n = 6 cilia. (***) p < 0.001. Scale bars, 5 μm.

The following figure supplements are available for figure 2:

**Figure supplement 1**. Whole-cell images corresponding to the images shown in *Figure 2B*.

**Figure supplement 2**. Whole-cell images corresponding to the images shown in *Figure 2E*.

**Figure supplement 3**. New D1R delivery to cilia increases over time.

**Figure supplement 4**. Control experiment for the ciliary delivery assay described in *Figure 2F*.

**Figure supplement 5**. Images of cilia from ciliary delivery assay.

**Figure supplement 6**. Bleaching control for the ciliary delivery assay.

**Figure supplement 7**. Bleed-through control for the ciliary delivery assay.

there was negligible bleed-through from the 488 channel into the 555 channel (*Figure 2—figure supplement 7*).

## Ciliary targeting of D1Rs is directed by the receptor's cytoplasmic tail

To begin to explore the biochemical mechanism of D1R ciliary targeting, we searched for structural determinants within the receptor that are required for ciliary localization. Previous studies of other cilia-localized receptors have identified targeting determinants located either in a cytoplasmic loop (*Berbari et al., 2008a*) or the cytoplasmic tail (C-tail; *Deretic et al., 1998*; *Corbit et al., 2005*; *Geng et al., 2006*; *Jenkins et al., 2006*). Progressive truncation of the D1R C-tail (*Figure 3A*) strongly reduced ciliary localization of receptors. Truncating the distal end had no effect (e.g., D1-415T; *Figure 3B*), but removing a larger portion strongly reduced ciliary receptor localization (D1-382T; *Figure 3C*). A 15-residue sequence within this region markedly reduced D1R ciliary localization when selectively deleted (D1Δ381–395; *Figure 3D*; whole-cell images for *Figure 3B–D* are shown in *Figure 3—figure supplement 1*). This effect was verified by both metrics of ciliary receptor targeting (*Figure 3E,F*), and the deletion did not disrupt overall surface expression of receptors (*Figure 3—figure supplement 2*).

We next asked if the D1R C-tail is sufficient to confer ciliary localization on a non-ciliary GPCR. Fusion of the entire D1R C-tail to DOR (*Figure 3G*) conferred robust ciliary localization. However,

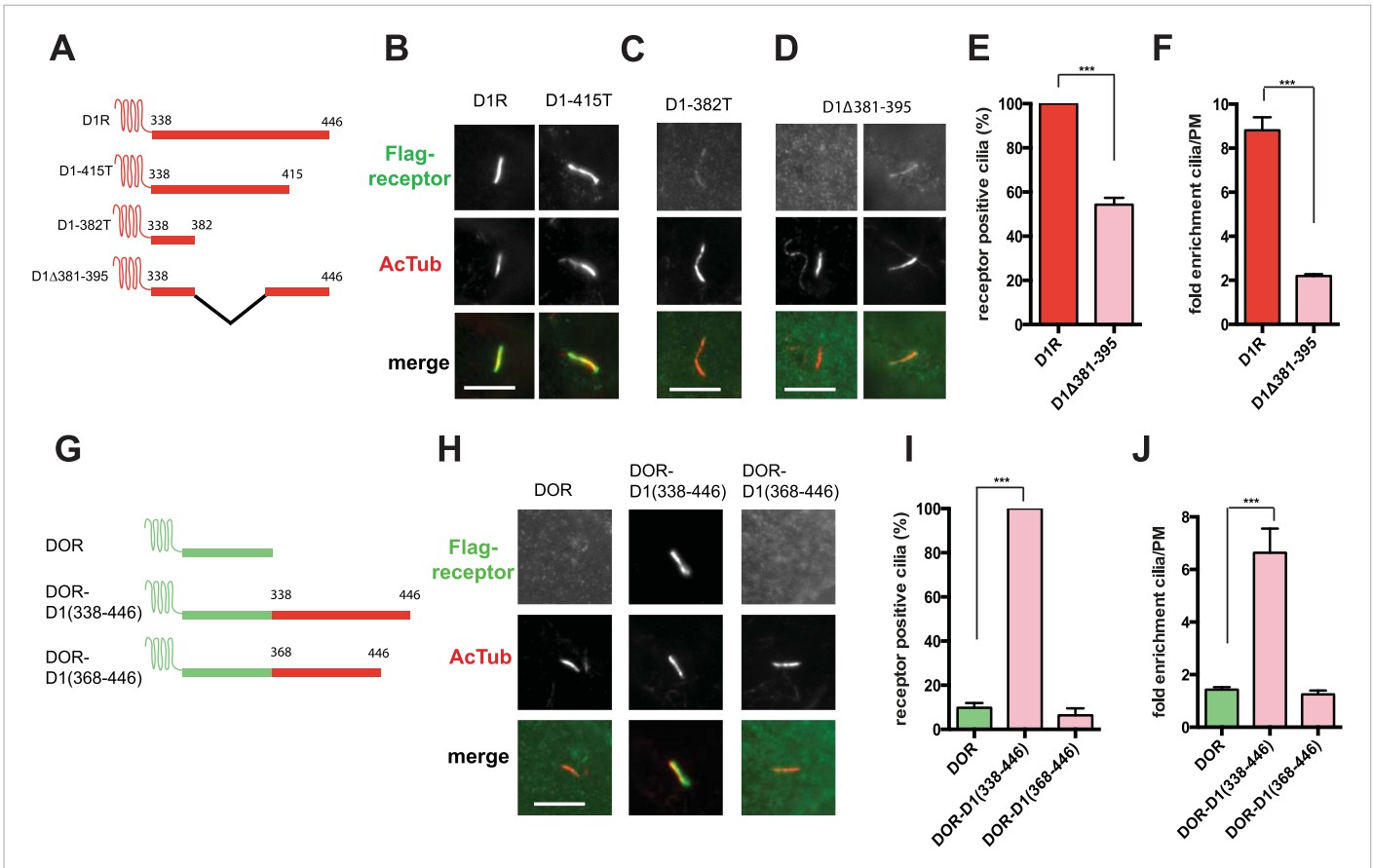

**Figure 3**. The D1R cytoplasmic tail is necessary and sufficient for ciliary receptor targeting. (**A**) Schematic representation of D1R C-tail mutations used in the present analysis. (**B**) Representative images of cells expressing Flag-tagged wild-type D1R (D1R) or a receptor construct truncated at residue 415 (D1-415T), showing robust ciliary localization of both. The merged image at bottom displays Flag-receptor in green and AcTub in red. (**C**) Representative image of a receptor construct truncated at residue 382 (D1-382T), showing near complete loss of receptor localization to cilia marked by AcTub. (**D**) Representative images of a D1R construct with internal deletion of residues 381–395 (D1Δ381-395), showing the range of phenotypes observed, from a complete loss of receptor localization in cilia to a pronounced reduction of ciliary receptor localization. (**E**) Quantification of the fraction of receptor (Flag)-positive cilia for Flag-tagged wild-type (D1R) or mutant (D1Δ381-395) receptor. The analysis is described in *Figure 1D*. (**F**) Quantification of the average fold-enrichment of wild-type (D1R) or mutant (D1Δ381-395) receptors on cilia. The analysis is described in *Figure 1E,F*. (**G**) Schematic representation of chimeric mutant receptors containing portions of the D1R cytoplasmic tail (in red) fused to the delta opioid peptide receptor (DOR) cytoplasmic tail (in green). (**H**) Representative images of cilia in cells expressing Flag-DOR, Flag-DOR-D1(338–446), or Flag-DOR-D1(368–446) showing that the D1R C-tail is sufficient to drive ciliary targeting of chimeric receptors. (**I**) Fraction of receptor (Flag)-positive cilia. (**J**) Average fold-enrichment of receptor (Flag) signal on cilia. Error bars represent SEM from n = 3 experiments with 10–20 cilia analyzed per experiment. (\*\*\*) p < 0.001. Scale bars, 5 µm.

The following figure supplements are available for figure 3:

**Figure supplement 1**. Whole-cell images corresponding to images shown in *Figure 3B–D*.

**Figure supplement 2**. Overall surface expression of D1R C-tail mutant.

**Figure supplement 3**. Whole-cell images corresponding to images shown in *Figure 3H*.

**Figure supplement 4**. Overall surface expression of DOR-derived constructs.

**Figure supplement 5**. The 15 residue sequence required for full ciliary targeting of D1R is not sufficient to confer ciliary localization on DOR.

fusion of a shorter fragment of the D1R C-tail was not sufficient to confer ciliary localization (*Figure 3H–J*; whole-cell images for *Figure 3H* shown in *Figure 3—figure supplement 3*), even though it contained the necessary 381–395 sequence, and chimeras exhibited similar levels of overall

surface expression (*Figure 3—figure supplement 4*). Additionally, we found that fusion of a smaller region of the D1R C-tail that also contains residues 381–395 (Flag-DOR-D1(379–400)) was not sufficient to confer ciliary localization on DOR. 23 out of 23 cells expressing the chimeric receptor had no visible enrichment of receptor in the cilium (*Figure 3—figure supplement 5*). Together, these results indicate that the structural information required for D1R ciliary targeting is contained in the receptor's C-tail and requires residues 381–395 for full activity. However, the targeting determinant is clearly not restricted to this 15-residue sequence, and it likely represents a more extended structure including residues in the proximal C-tail.

## Ciliary targeting of D1Rs is promoted by IFT-B complex proteins and KIF17

We next pursued a candidate-based RNA interference screen to search for trans-acting proteins required for ciliary D1R targeting (*Table 1*). Duplexes were transfected individually into IMCD3 cells stably expressing the Flag-D1R, using a clone with particularly strong ciliary receptor accumulation, and ciliary D1R localization was scored visually. No effect was found for siRNAs targeting several proteins implicated in ciliary localization other GPCRs, including TULP3 and BBSome components important for ciliary localization of SSTR3, MCHR1, and Gpr161 (*Table 1*; *Berbari et al., 2008b*; *Jin et al., 2010*; *Mukhopadhyay et al., 2010*, *2013*). Instead, two components of the IFT-B complex, IFT57 and IFT172, were identified (*Figure 4A*; whole-cell images shown in *Figure 4—figure supplement 1*) and knockdown was verified by qRT-PCR (*Figure 4—figure supplement 2*). IFT-B knockdown was complicated by reduced ciliogenesis (*Figure 4B*), but in the ciliated cells remaining in the transfected cell population, reduced D1R targeting was clearly evident. This was verified by both quantitative metrics of ciliary D1R localization, and this was specific to ciliary targeting because knockdown produced little or no effect on the overall surface expression of receptors (*Figure 4C,D*; *Figure 4—figure supplement 3*). As IFT57 tolerates an N-terminal epitope tag while IFT172 does not (*Follit et al., 2009*), we tested rescue of the IFT57 knockdown effect using an N-terminally HA-tagged IFT57 construct engineered with silent mutations in the sequence targeted by IFT57 siRNA (HA-IFT57-NTM). Verifying a specific requirement for IFT57 in ciliary D1R targeting, HA-IFT57-NTM restored

**Table 1**. siRNA knockdown screen

| Gene | Implication | +/− effect on D1 in cilia |
|---|---|---|
| TULP3 | SSTR3, MCHR1 cilia localization | − |
| Bbs4, Bbs2 | SSTR3, MCHR1 cilia localization | − |
| Arf4 | Rhodopsin localization to rod outer segment | − |
| Asap1 | Rhodopsin localization to rod outer segment | − |
| Kif7, Kif27 | Hedgehog signaling | − |
| Vps35 | Receptor trafficking | − |
| Rab15, Rab14, Rab8, Rab11 | Cilia associated Rabs | − |
| Rab4, Rab35 | Trafficking of receptors | − |
| Rab23 | Hedgehog signaling | + |
| Arl6 | Cilia associated small GTPase | − |
| IFT57 | Intraflagellar transport | + |
| IFT172 | Intraflagellar transport | + |
| Clathrin heavy chain | Receptor endocytosis | − |
| Pacs1 | Olfactory CNG channel cilia localization | − |
| Kif5c | Apical trafficking of cargo | − |
| Septin2 | Cilia diffusion barrier | − |

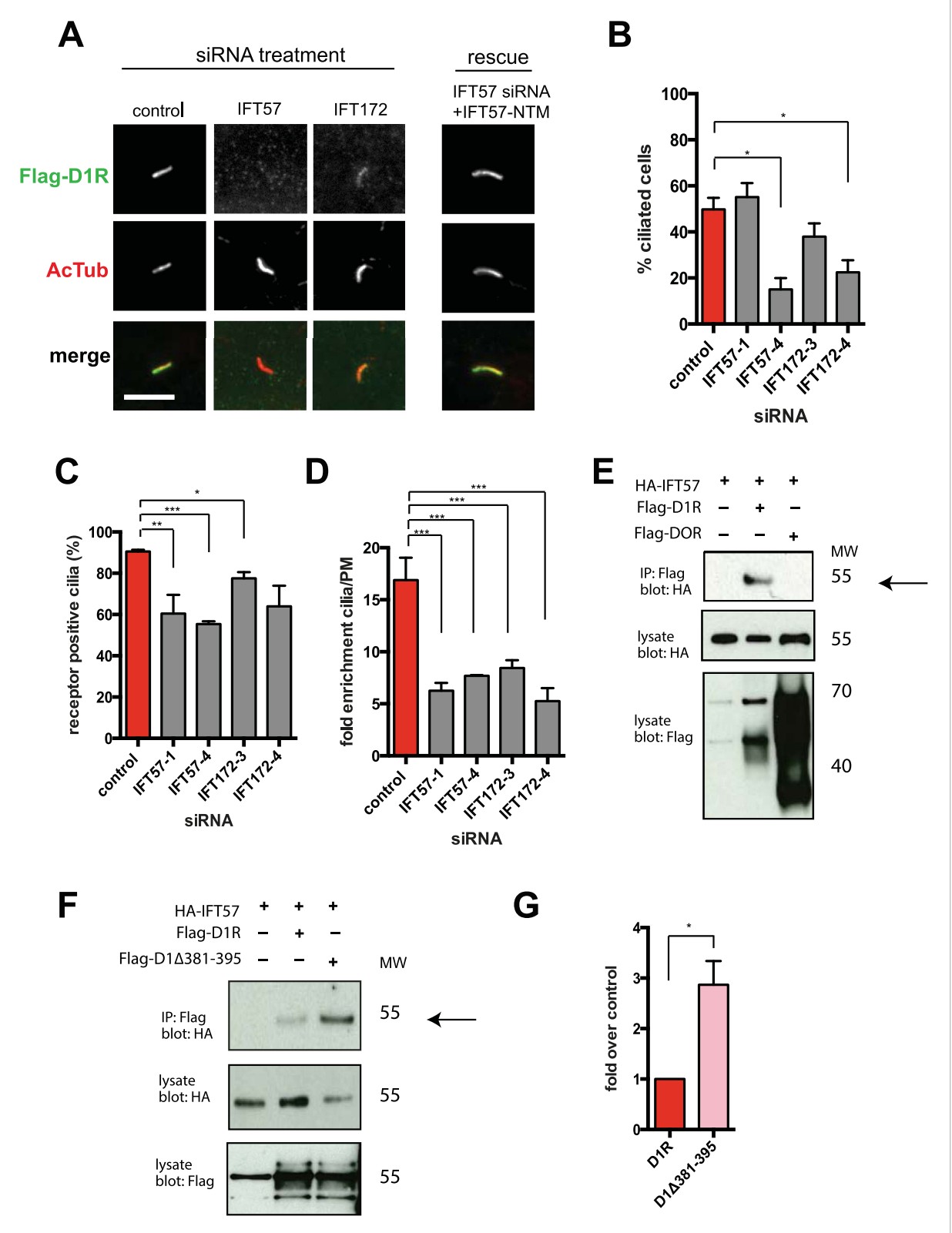

**Figure 4.** IFT-B complex proteins are necessary for D1R localization to cilia. (**A**) Representative images of cilia on cells stably transfected with Flag-D1R and transiently transfected with a non-silencing duplex (control) or siRNA targeting IFT57 (IFT57-1 and IFT57-4) or IFT172 (IFT172-3 and IFT172-4). In the merged image, Flag-D1R immunoreactivity is shown in green and AcTub in red. Duplexes targeting IFT57 and IFT172 caused a visually obvious reduction in ciliary D1R localization. The right column of images shows the rescue condition in which cells transfected with siRNA against IFT57 were additionally

*Figure 4. continued on next page*

*Figure 4. Continued*

transfected with IFT57 that is not targetable by the IFT57 siRNA (HA-IFT57-NTM). Scale bar, 5 µm. (**B**) Effect of siRNAs on the fraction of cells in the population possessing a visible cilium marked by AcTub. Error bars represent SEM from 150 cells counted in n = 3 independent experiments. (**C**) Fraction of D1R (Flag)-positive cilia. Error bars represent SEM for n = 3 experiments with 50 cells counted in each experiment. (**D**) Average fold-enrichment of D1R (Flag) signal on cilia. Error bars represent SEM for n = 3 experiments with 15–40 cilia analyzed per experiment. (**E**) Association of IFT57 with the D1R but not DOR demonstrated by co-immunoprecipitation. Cells were transfected with the constructs indicated above each lane. Cell extracts were blotted for HA and Flag; HA-IFT57 resolved as a sharp band at its expected apparent molecular mass and Flag-tagged receptors resolved as heterogeneous species consistent with complex glycosylation as shown previously. Specific co-immunoprecipitation is indicated by HA-IFT57 detected in the Flag-D1R pull-down but not in Flag-DOR pull-down. Molecular mass markers (in kDa) are shown on right side of blots. The results in panel **E** are representative of n = 3 independent experiments. (**F**) Increased association of IFT57 with D1Δ381-395 relative to D1R demonstrated by co-IP. Cells were transfected with the constructs indicated above each lane. Cell extracts were blotted for HA and Flag. More HA-IFT57 was detected in the Flag-D1Δ381-395 pull-down than the Flag-D1R pull-down. Molecular mass markers (in kDa) are shown on right side of blots. The results in panel **E** are representative of n = 3 independent experiments. (**G**) Immunoblots from multiple experiments were scanned in the linear range, as described in 'Materials and methods', to estimate the amount of IFT57 co-IPed with the indicated receptors. Expressed as a fold increase over control where control is D1R. Error bars represent SD from n = 3 experiments. (*) $p < 0.05$; (**) $p < 0.01$; (***) $p < 0.001$.

The following figure supplements are available for figure 4:

**Figure supplement 1**. Whole-cell images corresponding to the images shown in *Figure 4A*.

**Figure supplement 2**. Verification of IFT-B knockdown.

**Figure supplement 3**. IFT-B knockdown has little effect on overall surface receptor expression.

normal ciliary D1R localization in 30 out of 30 cells analyzed (*Figure 4A*, rescue; whole-cell images verifying HA-IFT57-NTM expression in *Figure 4—figure supplement 1*).

Co-immunoprecipitation analysis detected physical association of HA-IFT57 with Flag-D1R, and this was specific because HA-IFT57 did not detectably co-IP with Flag-DOR even when the latter was expressed at higher levels than Flag-D1R (*Figure 4E*). To ask if IFT57 interaction with D1R is directly congruent with ciliary targeting activity, we carried out co-immunoprecipitation analysis comparing HA-IFT57 pull-down with wild-type Flag-D1R and FlagD1Δ381-395. Deletion of resides 381–395 of the D1R C-tail, a mutation that profoundly impairs ciliary targeting, did not reduce IFT57 co-IP. To the contrary, the co-IP signal was significantly enhanced (*Figure 4F,G*). This suggests that the D1R C-tail, while both necessary and sufficient for ciliary receptor targeting, likely functions in a more complex manner than can be explained by a single interaction surface with IFT-B.

IFT-B is known to associate with KIF3A (kinesin-II, a heterotrimeric kinesin-2) and KIF17 (a homodimeric kinesin-2), plus end-directed kinesins that mediate anterograde cargo movement toward or within cilia (reviewed in *Pedersen and Rosenbaum, 2008*). KIF3A is required for delivery of axonemal components and overall ciliogenesis, while KIF17 is not essential for ciliogenesis and is proposed to have more specialized cargo transport functions (*Jenkins et al., 2006*; *Zhao et al., 2012*). We verified the presence of KIF17-IFT57 complexes in IMCD3 cells, as reported previously by others (*Insinna et al., 2008*; *Howard et al., 2013*), by co-immunoprecipitation (*Figure 5A*). Thus, we hypothesized that KIF17 may function as a specific motor supporting D1R delivery to cilia. To test this, we introduced a point mutation in the KIF17 motor domain at a conserved residue in the switch II region (KIF17-G234A) that is essential for kinesin motor activity (*Figure 5—figure supplement 1*; *Rice et al., 1999*). D1Rs were visible on cilia in essentially all cells expressing the motor-defective HA-KIF17-G234A, but the degree of ciliary receptor enrichment was greatly reduced (*Figure 5B*). In contrast, expression of wild-type KIF17 did not visibly affect D1R ciliary localization (*Figure 5B*; whole-cell images verifying HA-KIF17-G234A and HA-KIF17 expression shown in *Figure 5—figure supplement 2*). Verifying this, KIF17-G234A, but not KIF17, selectively reduced the fold-enrichment metric while having little effect on the fraction of receptor-positive cilia (*Figure 5C,D*). As an independent assessment of the role of KIF17 on D1R ciliary localization, we examined the effect of expressing a different dominant negative KIF17 construct (HA-KIF17-DN) containing only the cargo-binding domain (*Jenkins et al., 2006*). Expression of HA-KIF17-DN also significantly reduced D1R ciliary enrichment (*Figure 5—figure supplement 3*). Further, this effect was specific to D1Rs because

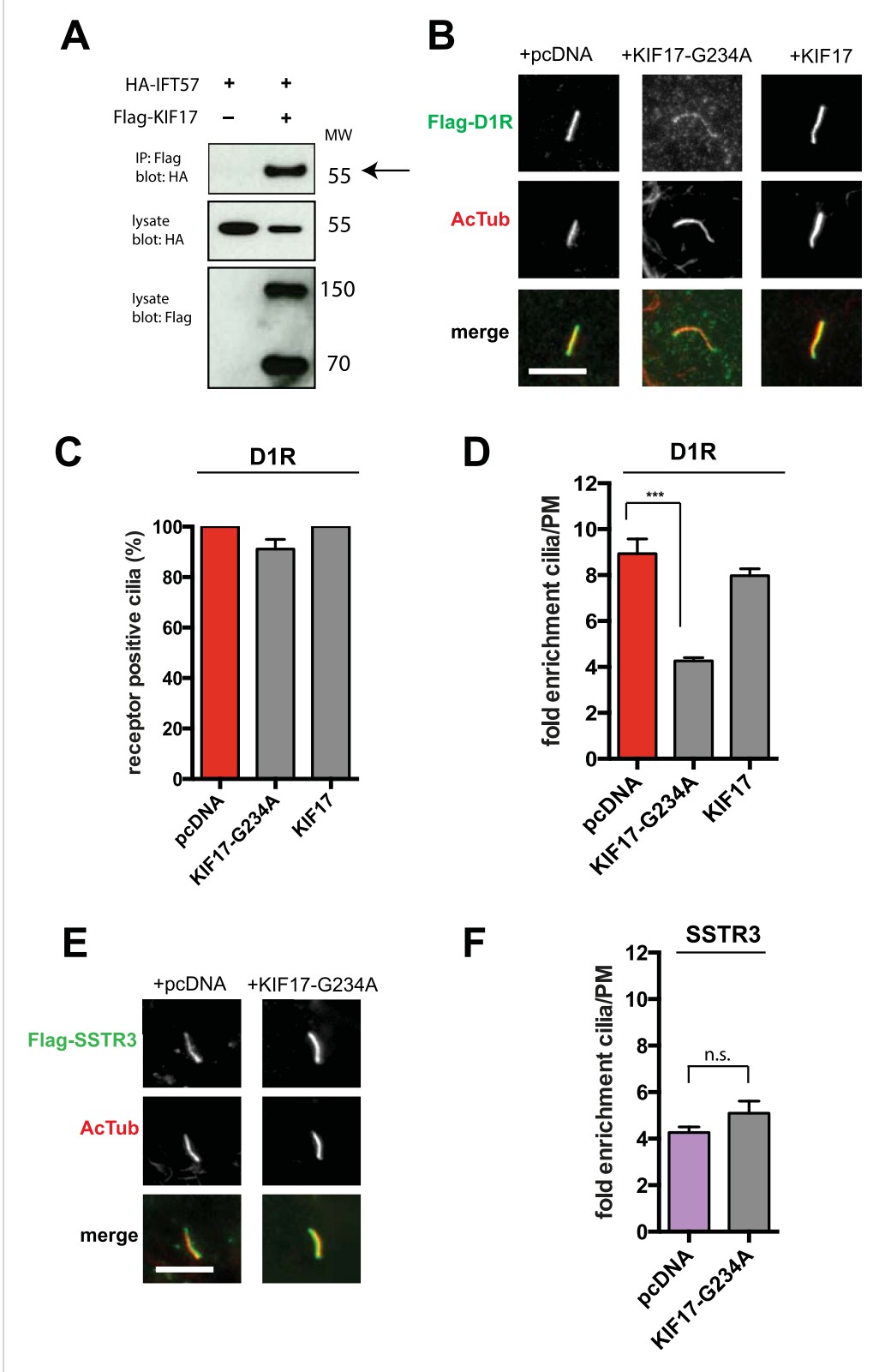

**Figure 5**. KIF17 motor activity is required for full D1R enrichment in cilia. (**A**) Association of IFT57 with KIF17 indicated by co-immunoprecipitation. Cells were transfected with expression constructs indicated at top of the panel, and extracts were blotted for HA to detect IFT57 and Flag to detect KIF17. HA-IFT57 resolved as expected and described in *Figure 4E*. KIF17 resolved as two species with the top band corresponding to the expected

*Figure 5. continued on next page*

*Figure 5. Continued*

molecular mass of the full-length protein. Specific co-immunoprecipitation is indicated by HA-IFT57 detected in the Flag pull-down from cells expressing Flag-KIF17 pull-down (arrow) but not from cells in which Flag-KIF17 was not expressed. Molecular mass markers (in kDa) shown on right. (**B**) Representative images of cilia on cells co-transfected with Flag-D1R and control empty vector (+pcDNA), a plasmid encoding HA-tagged KIF17 (+KIF17), or a plasmid encoding an HA-tagged KIF17 construct harboring a point mutation in a conserved residue that disrupts kinesin motor activity (+KIF17-G234A). Robust ciliary localization of Flag-D1R was observed in cells expressing control plasmid or the wild-type KIF17 construct, but markedly reduced ciliary enrichment of D1Rs was observed in cells expressing motor-defective mutant KIF17. (**C**) Quantification of the effect of disrupting KIF17 motor activity on the fraction of D1R (Flag)-positive cilia. (**D**) Quantification of the effect of disrupting KIF17 motor activity on average fold-enrichment of D1R (Flag) on cilia. Disrupting KIF17 motor activity strongly reduced the degree of D1R enrichment on the ciliary membrane without blocking D1R access to cilia. (**E**) Representative images of cilia on cells co-transfected with Flag-SSTR3 and with control empty vector (+pcDNA) or a plasmid encoding motor domain-mutant KIF17 (+KIF17-G234A). Disrupting KIF17 motor activity did not detectably affect Flag-SSTR3 localization to cilia. (**F**) Quantification of the effect of disrupting KIF17 motor activity on average fold-enrichment of somatostatin-3 receptor (SSTR3) (Flag) on cilia. Disrupting KIF17 motor activity did not detectably affect ciliary enrichment of SSTR3. Error bars represent SEM from n = 3 independent experiments with 10–20 cilia analyzed in each experiment. (***) p < 0.001. Scale bars, 5 µm.

The following figure supplements are available for figure 5:

**Figure supplement 1**. Switch II mutation in KIF17.

**Figure supplement 2**. Whole-cell images corresponding to the images shown in *Figure 5B*.

**Figure supplement 3**. Independent verification of KIF17 requirement for D1R ciliary enrichment.

**Figure supplement 4**. Whole-cell images corresponding to the images shown in *Figure 5E*.

**Figure supplement 5**. The KIF17 motor domain mutation has little effect on overall surface expression of receptors.

KIF17-G234A did not reduce ciliary enrichment of SSTR3 (*Figure 5E,F*; whole-cell images verifying HA-KIF17-G234A expression in *Figure 5—figure supplement 4*). Expression of HA-KIF17-G234A did not significantly affect overall surface expression of either D1R or SSTR3 (*Figure 5—figure supplement 5*).

## A discrete and essential function of Rab23 in the ciliary targeting mechanism

Our siRNA screen also identified Rab23 as a candidate whose knockdown caused a pronounced reduction of D1R localization to cilia (*Figure 6A*; whole-cell images in *Figure 6—figure supplement 1*) without affecting ciliogenesis (*Figure 6B*). Knockdown was verified by qRT-PCR (*Figure 6—figure supplement 2*). Rab23 knockdown strongly reduced both quantitative metrics of ciliary D1R targeting, without affecting overall surface expression of receptors (*Figure 6C,D*; *Figure 6—figure supplement 3*). Rab23 knockdown also blocked ciliary targeting activity of the D1R C-tail assessed through fusion to the normally cilia-excluded DOR (*Figure 6E,F*). This was unexpected because Rab23 was not known previously to be required for ciliary targeting of any signaling receptor or other membrane cargo.

To ask if Rab23 affects additional receptor cargoes, we carried out the same experiment investigating ciliary localization of SSTR3. Duplexes were transfected individually into IMCD3 cells stably expressing SSTR3-GFP. Rab23 knockdown significantly reduced ciliary targeting of SSTR3, as assessed by epifluorescence microscopy and both quantitative metrics (*Figure 6G–I*; whole-cell images shown in *Figure 6—figure supplement 4*).

Rab23 was not only necessary for ciliary localization of wild-type D1Rs and the ciliary targeting activity of the D1R C-tail, but it was also sufficient to rescue ciliary targeting of targeting-defective D1Rs when fused to the C-tail. Fusing wild-type Rab23 to the C-tail of the ciliary localization-defective D1Δ381-395 mutant receptor (D1Δ381-395-Rab23, *Figure 7A*) rescued robust ciliary targeting

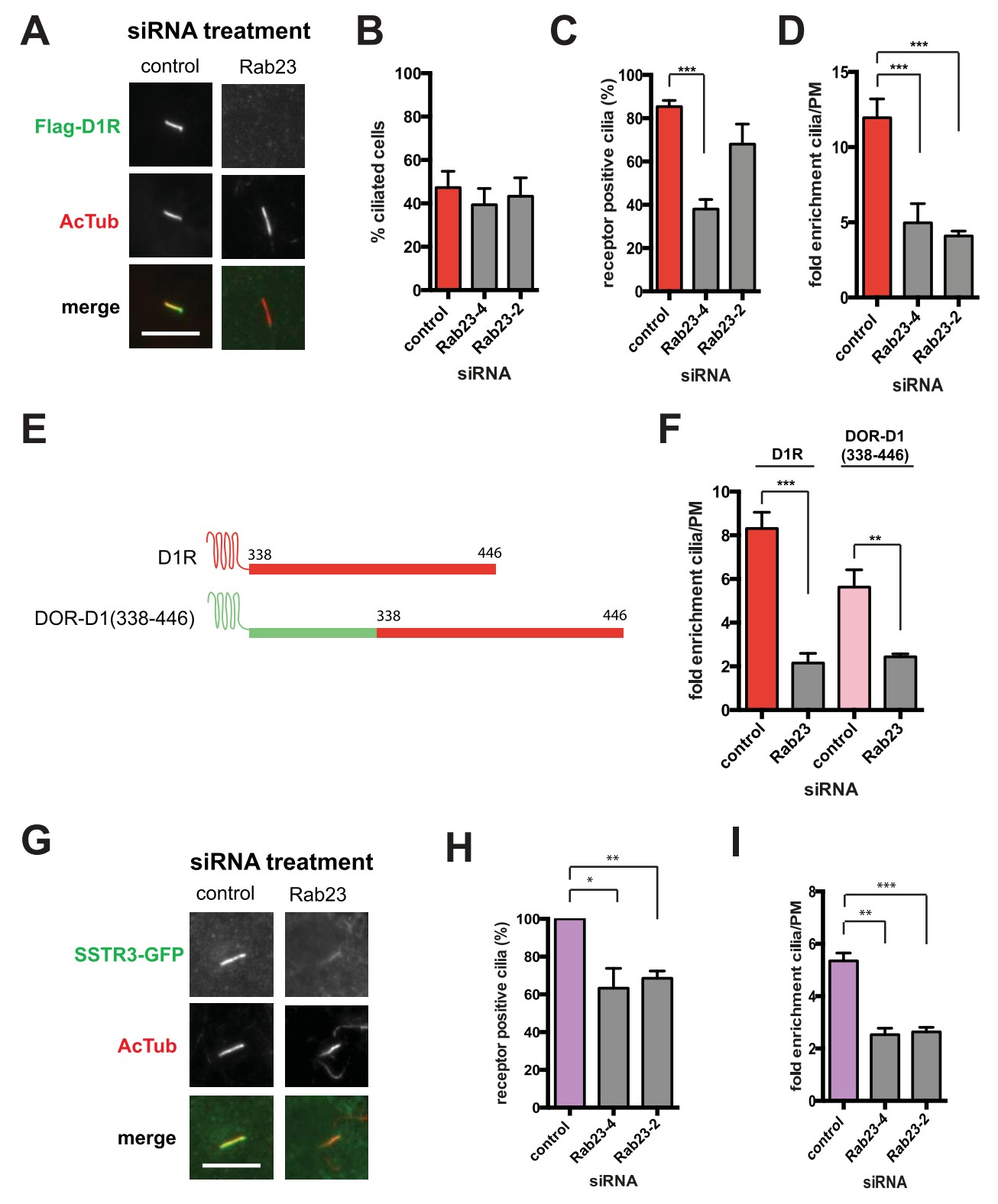

**Figure 6**. Rab23 is necessary for D1R localization to cilia. (**A**) Representative images of cilia on cells stably transfected with Flag-D1R and transiently transfected with a non-silencing duplex (control) or siRNA targeting Rab23 (Rab23-4 and Rab23-2). Rab23 knockdown strongly reduced Flag-D1R localization to cilia. (**B**) Quantification of the siRNA effect on the fraction of cells in the population possessing a visible cilium marked by AcTub. Error bars represent SEM from 150 cells counted in n = 3 experiments. (**C**) Quantification of the Rab23 knockdown effect on the fraction of D1R (Flag)-positive cilia. *Figure 6. continued on next page*

*Figure 6. Continued*

Error bars represent SEM from n = 3 experiments with 50 cells counted per experiment. (**D**) Quantification of the Rab23 knockdown effect on average fold-enrichment of D1R (Flag) signal on cilia. Error bars represent SEM from n = 3 independent experiments with 20–30 cilia analyzed per experiment. (**E**) Schematic representation of wild-type D1R and the cilia-targeted DOR-D1(338–446) chimeric mutant receptor (duplicated from *Figure 3G*). (**F**) Quantification of the Rab23 knockdown effect on average fold-enrichment of Flag-D1R (D1R) and the Flag-tagged chimeric mutant receptor (DOR-D1 (338–446)) on cilia of transiently transfected cells. Error bars represent SEM from n = 3 independent experiments with 10–20 cilia analyzed per experiment. (**G**) Representative images of cilia on cells stably transfected with SSTR3-GFP and transiently transfected with a non-silencing duplex (control) or siRNA targeting Rab23 (Rab23-4 and Rab23-2). Rab23 knockdown strongly reduced SSTR3-GFP localization to cilia. (**H**) Quantification of the Rab23 knockdown effect on the fraction of SSTR3-GFP (GFP) positive cilia. Error bars represent SEM from n = 3 experiments with 50 cells counted per experiment. (**I**) Quantification of the Rab23 knockdown effect on average fold-enrichment of SSTR3 (GFP) signal on cilia. Error bars represent SEM from n = 3 experiments with 10–20 cilia analyzed in each experiment. (*) p < 0.05; (**) p < 0.01; (***) p < 0.001. Scale bars, 5 μm.

The following figure supplements are available for figure 6:

**Figure supplement 1**. Whole-cell images for corresponding images shown in *Figure 6A*.

**Figure supplement 2**. Verification of Rab23 knockdown.

**Figure supplement 3**. Rab23 knockdown has little effect on overall surface receptor expression.

**Figure supplement 4**. Whole-cell images for corresponding images shown in *Figure 6G*.

(*Figure 7B*; whole-cell images shown in *Figure 7—figure supplement 1*). This effect was dependent on the nucleotide state of Rab23 because a GTP binding-defective mutant allele (D1Δ381-395-Rab23-S23N) failed to produce detectable ciliary receptor localization, while fusion of an activated Rab23 allele (D1Δ381-395-Rab23-Q68L) drove ciliary localization even more robustly than the D1R C-tail itself (*Figure 7B–D*; overall surface expression is shown in *Figure 7—figure supplement 2*). Rab23 was detectable but not highly concentrated in cilia (*Figure 7E*). Therefore, we do not think Rab23 fusion confers ciliary localization on receptors by simple tethering.

To ask if Rab23 fusion is fully sufficient to promote ciliary targeting of receptors, we carried out similar experiments fusing Rab23-Q68L to DOR, which is normally undetectable on cilia. Remarkably, activated Rab23 drove robust ciliary localization of DOR (DOR-Rab23-Q68L, *Figure 7F–I*; whole-cell images are shown in *Figure 7—figure supplement 3*). While fusion of Rab23-Q68L to the mutant D1R increased overall receptor surface expression (*Figure 7—figure supplement 2*), the opposite was observed for fusion to DOR (*Figure 7—figure supplement 4*). This indicates that the ciliary targeting activity of Rab23 is separable from its effects on overall surface receptor expression. Moreover, this ciliary targeting activity was specific for Rab23 because fusion of an activated allele of the closely related Rab paralogue, Rab11 (DOR-Rab11-Q70L), or of Rab8 (DOR-Rab8-Q67L) that is known to localize to cilia, failed to drive detectable ciliary targeting (*Figure 7F,G*). The ability of activated Rab23 to confer ciliary localization on a non-ciliary GPCR was not limited to DOR. Fusion of activated Rab23 to the C-tails of the mu-opioid receptor (MOR) and beta-2-adrenergic receptor (B2AR), which are normally excluded from cilia, conferred robust ciliary localization on both receptors (*Figure 7—figure supplement 5*). Together, these results support the hypothesis that Rab23 plays a key role in determining the specificity of ciliary receptor targeting.

We next sought to investigate if the identified protein components required for D1R ciliary targeting function in an integrated pathway. As noted above, disrupting KIF17 motor activity strongly reduced ciliary enrichment of the wild-type D1R. In contrast, ciliary enrichment driven by direct fusion of the activated Rab23 to the D1R (D1Δ381-395-Rab23-Q68L) was unaffected by this manipulation (*Figure 8A–C*; whole-cell images verifying HA-KIF17-G234A expression are shown in *Figure 8—figure supplement 1*). Additionally, full ciliary enrichment of D1Δ381-395-Rab23-Q68L remained in the presence of IFT172 knockdown (*Figure 8D*), suggesting that fusion to activated Rab23 can also override the IFT-B requirement. We also noted that Rab23 knockdown did not prevent or reduce D1R association with IFT-B, as estimated by co-immunoprecipitation of IFT57 (*Figure 8E*). To the contrary, Rab23 knockdown tended to increase the IFT57-D1R co-IP signal (*Figure 8F*). In contrast to its clear effect on the ciliary concentration of D1Rs, disrupting KIF17 motor activity did

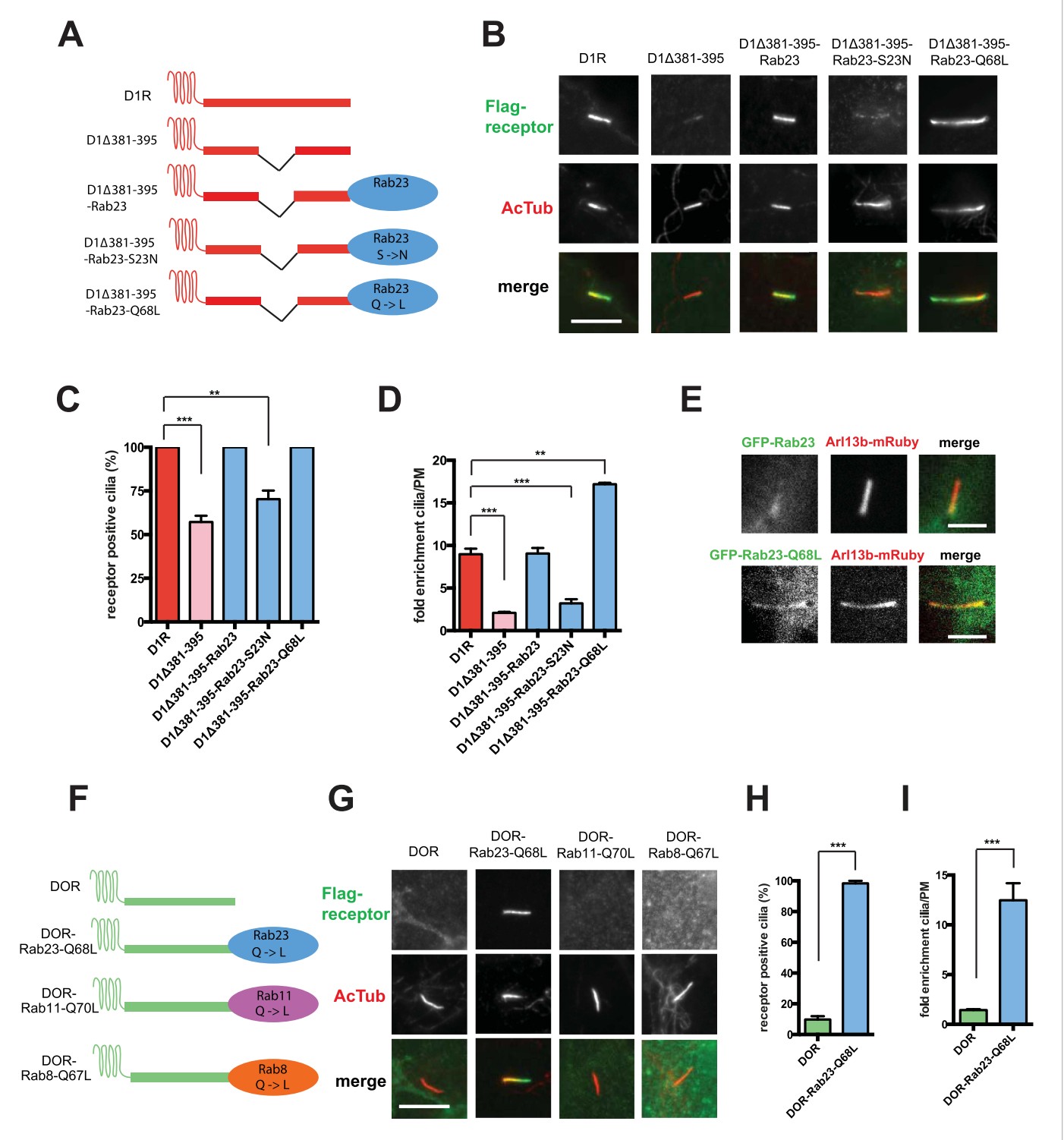

**Figure 7**. Rab23 is sufficient to drive ciliary localization of a non-ciliary GPCR. (**A**) Schematic representation of the D1R-derived constructs examined. D1R-derived sequence is depicted in red and Rab23 sequence in blue. Flag-tagged wild-type D1R was compared to the ciliary targeting-impaired mutant D1R (D1Δ381-395), and to the ciliary targeting-impaired mutant D1R fused to wild-type Rab23 (D1Δ381-395-Rab23), inactive mutant Rab23 (D1Δ381-395-Rab23-S23N) or activated mutant Rab23 (D1Δ381-395-Rab23-Q68L). (**B**) Representative images of cilia on cells transiently expressing Flag-tagged versions of the indicated receptor constructs. Fusion of either wild-type or activated Rab23 to the cilia targeting-defective D1R visibly enhanced ciliary localization of receptors. (**C**) Quantification of the fraction of receptor (Flag)-positive cilia. (**D**) Average fold-enrichment of receptor (Flag) on cilia. (**E**) Representative live-cell images of GFP-tagged Rab23 or Rab23-Q68L localization relative to cilia marked by Arl13b-mRuby after transient co-transfection. (**F**) Schematic

*Figure 7. continued on next page*

*Figure 7. Continued*

representation of the DOR-Rab fusions. DOR-derived sequence is depicted in green, Rab23 in blue, Rab11 in violet, and Rab8 in orange. (**G**) Representative images of cilia on cells transiently expressing Flag-tagged versions of the indicated receptor constructs. Wild-type DOR was not detected on cilia. Fusion of activated (Q68L) Rab23 produced strong ciliary localization, while fusion of activated (Q70L) Rab11 or activated (Q67L) Rab8 failed to do so. (**H**) Fraction of receptor (Flag)-positive cilia. (**I**) Average fold-enrichment of receptor (Flag) signal on cilia. Error bars represent SEM from n = 3 independent experiments with 10–20 cilia analyzed in each experiment. (**) p < 0.01; (***) p < 0.001. Scale bars, 5 μm. GPCR, G protein-coupled receptor.

The following figure supplements are available for figure 7:

**Figure supplement 1**. Whole-cell images corresponding to images shown in *Figure 7B*.

**Figure supplement 2**. Overall surface expression of D1R-Rab23 fusion constructs.

**Figure supplement 3**. Whole-cell images corresponding to images shown in *Figure 7G*.

**Figure supplement 4**. Overall surface expression of DOR-Rab23 fusion.

**Figure supplement 5**. Rab23 is sufficient to drive ciliary localization of several non-ciliary GPCRs.

not prevent Rab23 localization to cilia (*Figure 8—figure supplement 2*). Together, these results suggest that IFT-B/KIF17 and Rab23 are all required for efficient targeting of D1Rs to cilia and may indeed function in an integrated pathway.

## Discussion

The physiological signaling functions of primary cilia critically depend on the specificity with which particular receptors are targeted to the ciliary membrane (*Corbit et al., 2005*; *Garcia-Gonzalo and Reiter, 2012*). We identified a discrete mechanism of ciliary receptor targeting through study of the D1R.

Past work has focused primarily on receptor delivery from a post-Golgi membrane source. In contrast, we found that D1Rs are delivered to cilia from the extra-ciliary plasma membrane. We also found that ciliary D1R delivery is directed by the receptor C-tail. However, we were unable to find any sequence in the D1R C-tail conforming to a previously defined ciliary targeting motif (*Deretic et al., 1998*; *Geng et al., 2006*; *Jenkins et al., 2006*; *Berbari et al., 2008a*). Also, our mutational studies suggest that the structural determinant required for ciliary D1R targeting is a relatively extended structure. We identified a distinct set of trans-acting proteins important for ciliary targeting of D1Rs. We were unable to detect a requirement for TULP3 or BBSome components, although these are essential for ciliary localization of SSTR3 and MCHR1 (*Berbari et al., 2008b*; *Jin et al., 2010*; *Mukhopadhyay et al., 2010*). We also did not detect a requirement for Arf4, ASAP1, or Rab11, which are essential for rhodopsin delivery to the rod outer segment (*Deretic et al., 2005*; *Wang et al., 2012*). Thus, D1Rs add to a growing appreciation that there exist receptor-specific differences in mechanisms of ciliary membrane targeting.

Nevertheless, some similarities are evident. The present results are consistent with IFT-B functioning in anterograde cargo delivery to the cilium (*Silverman and Leroux, 2009*; *Garcia-Gonzalo and Reiter, 2012*). Several IFT-B components (IFT20, IFT57, and IFT140) have been implicated in rhodopsin delivery to the rod outer segment (*Krock and Perkins, 2008*; *Keady et al., 2011*; *Crouse et al., 2014*), and IFT172 was recently implicated in Smo localization to primary cilia (*Kuzhandaivel et al., 2014*). The present finding that IFT57 and IFT172 are required for ciliary D1R targeting provides further evidence that IFT-B contributes to ciliary delivery of select membrane cargoes.

Our findings are also consistent with KIF17 functioning as an ancillary motor supporting ciliary cargo delivery. KIF17 is necessary for ciliary localization of an olfactory cyclic nucleotide-gated ion channel (*Jenkins et al., 2006*), but to our knowledge, KIF17 has not been shown previously to function in localizing a GPCR to cilia. Our results identify a specific role of KIF17, and of KIF17 motor activity, in

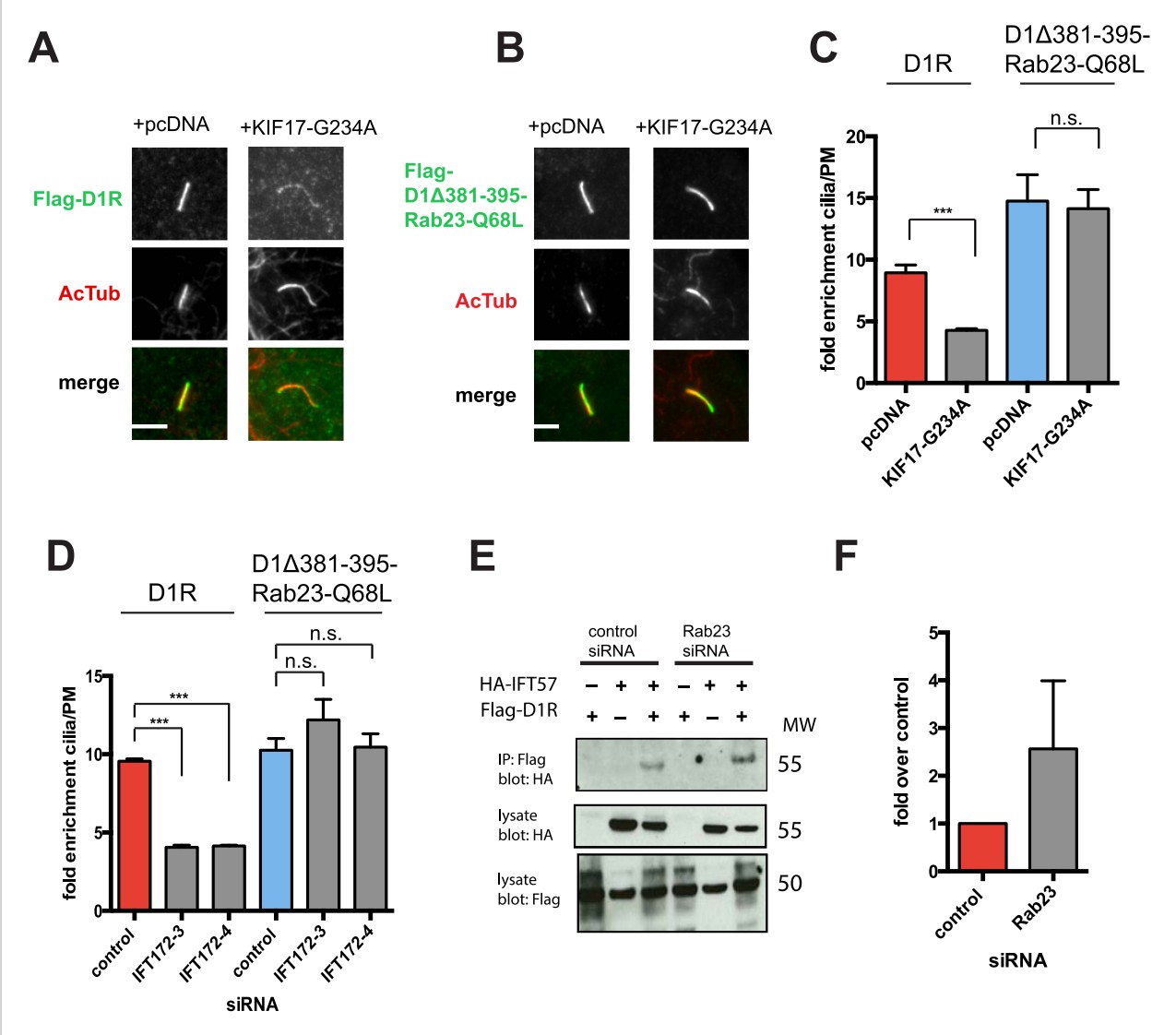

**Figure 8**. Evidence IFT-B, KIF17, and Rab23 function in an integrated ciliary delivery pathway. (**A**) Representative images showing the effect of motor domain-mutant KIF17 (+KIF17-G234A) on Flag-tagged wild-type D1R localization to the cilium (from *Figure 5B*). (**B**) Representative images from an identical experiment examining localization of the Flag-tagged D1R fusion to activated Rab23 (D1Δ381-395-Rab23-Q68L). Scale bars, 5 µm. (**C**) Average fold-enrichment of D1R and D1Δ381-395-Rab23-Q68L (Flag) signal on cilia. Wild-type D1R localization to cilia was strongly reduced by motor-defective KIF17, but direct Rab23 fusion effectively bypassed this requirement. (**D**) Effect of IFT172 knockdown on average fold-enrichment of D1R and D1Δ381-395-Rab23-Q68L (Flag) signal on cilia. Wild-type D1R localization to cilia was strongly reduced by IFT172 knockdown, but direct Rab23 fusion effectively bypassed this requirement. Error bars represent SEM from n = 3 independent experiments with 10–20 cilia analyzed in each experiment. (***) p < 0.001. (**E**) Co-immunoprecipitation analysis showing that Rab23 is not necessary for D1R association with IFT57. The analysis and presentation of data are described in *Figure 4E*. (**F**) Immunoblots from multiple experiments were quantified in the linear range to estimate the amount of IFT57 co-IPed. The result is expressed as a fold-increase over the control siRNA condition. Error bars represent SD from n = 3 experiments.

The following figure supplements are available for figure 8:

**Figure supplement 1**. Whole-cell images corresponding to the images shown in *Figure 8B*.

**Figure supplement 2**. Disruption of KIF17 motor activity does not affect Rab23 ciliary localization.

promoting ciliary concentration of the D1R but not the SSTR3. Further supporting the proposed role of KIF17 as an anterograde transport motor for D1Rs, we found that D1Rs specifically co-IP IFT57 from intact cells (*Figure 4E*) and verified IFT57 association with KIF17 (*Figure 5A*; *Insinna et al., 2008*;

*Howard et al., 2013*). An interesting related observation is that deletion of residues 381–395 in the D1R tail, a manipulation that inhibits D1R ciliary targeting, did not disrupt the D1R-IFT57 interaction, but rather, increased it. This supports the idea that ciliary targeting of D1Rs is likely a complex process.

A remarkable finding in the present study is that Rab23 is an essential component of the D1R ciliary targeting mechanism. To our knowledge, this represents the first evidence that Rab23 is required for ciliary localization of any signaling receptor or membrane cargo. Previous studies of other cilia-localized membrane proteins, such as polycystin-2 and Smo, have not observed such a requirement (*Eggenschwiler et al., 2006*; *Boehlke et al., 2010*; *Hoffmeister et al., 2011*). Thus, we think it likely that Rab23's function in ciliary membrane targeting is specific to a subset of cilia-localized cargoes. We note that ciliary localization of SSTR3 is also sensitive to Rab23 knockdown, even though, in contrast to D1R, its ciliary targeting is insensitive to manipulation of KIF17 motor activity. This provides further support for the existence of receptor-specific differences in the ciliary targeting mechanism.

Altogether, the present findings support the conclusion that D1Rs are targeted to the cilium from the extra-ciliary plasma membrane through a complex mechanism involving IFT-B, KIF17, and Rab23. Our results support the hypothesis that these components function together in an integrated ciliary delivery pathway and suggest that they have distinguishable functional effects in the delivery pathway. First, disrupting KIF17 motor activity strongly reduced ciliary D1R enrichment without affecting the fraction of receptor-positive cilia. This is potentially consistent with KIF17 motor activity promoting ciliary D1R concentration, but not being essential for ciliary D1R access. Second, direct fusion of activated Rab23 to the D1R C-tail can promote ciliary targeting in a manner that is apparently insensitive to KIF17 motor activity. Third, Rab23 knockdown did not disrupt the D1R/IFT-B interaction as estimated by co-immunoprecipitation. These results also support the idea that the lateral delivery mechanism is complex, but additional studies are required to further delineate the precise functions of each of these identified components.

The present results raise a number of interesting questions for future study. First, given that receptor accumulation in the ciliary membrane is dependent on Rab23 nucleotide state, an important next question is how this nucleotide state is controlled. A second question, which may be related to the first, is how selective cargo engagement with the ciliary delivery mechanism is determined. The present data strongly suggest a functional interaction between Rab23 and the D1R ciliary targeting determinant, but we have been unable, so far, to establish direct physical interaction between the D1R and Rab23. It is conceivable that IFT-B links D1Rs and Rab23, or that unidentified additional protein(s) explain the functional interaction observed. A third question is to ascertain precisely how Rab23 directs receptor delivery to the ciliary membrane. Based on precedent of other Rab protein functions, we speculate that there is a specific effector of Rab23 that operates at or near the ciliary diffusion barrier. A fourth question is what broader physiological significance the discrete, Rab23-dependent ciliary targeting mechanism has. We note that a mutation in Rab23 produces excessive Hedgehog signaling in vivo (*Eggenschwiler et al., 2001*). One possibility is that this reflects a disruption of ciliary signaling normally mediated by a Rab23-dependent receptor. Thus, further investigation of the receptor-specific ciliary targeting mechanism identified here may provide fundamental insight into the role of primary cilia as physiological signaling devices and toward understanding pathologies associated with ciliary defects.

## Materials and methods

### Cell culture and transfection

IMCD3 cells (ATCC) were grown in DMEM/Ham's F-12 Medium supplemented with 10% fetal bovine serum (UCSF Cell Culture Facility, San Francisco, California, USA).

Flag-D1R, Flag-DOR, Flag-MOR, and Flag-B2AR constructs were described previously (*Vickery and von Zastrow, 1999*; *Tanowitz and Von Zastrow, 2003*; *Yu et al., 2010*). KIF17 and KIF17-DN cDNA (*Jenkins et al., 2006*) was a gift from Kristen Verhey (University of Michigan, Ann Arbor). 2XHA-KIF17 was created using PCR and ligation into pIRES. HA-KIF17-DN was created using PCR and ligation into pIRES. 2XHA-KIF17-G234A was generated using site-directed mutagenesis (Phusion Site-Directed Mutagenesis Kit, Thermo Scientific, Waltham, MA). SSTR3-GFP IMCD3 stable cells were a gift from Maxence Nachury (Stanford University). IFT57 cDNA was a gift from Wallace Marshall (UCSF, San Francisco). HA-IFT57 was created using PCR and ligation into pIRES. HA-IFT57-NTM was generated

using site-directed mutagenesis (Phusion Site-Directed Mutagenesis Kit, Thermo Scientific) of HA-IFT57 to change the site targeted by IFT57-4 siRNA using primers 5′-GTCACCCCAGAGTCTGCGA TAGGGTTCTACTAAACACGTGGGCTTCC-3′ and 5′-AGCTGCATGCATGTCCCTGGTCATGTTGC-3′.

Flag-tagged D1-415T, D1-382T, and D1Δ381-395 were created by site-directed mutagenesis (Phusion Site-Directed Mutagenesis Kit, Thermo Scientific). Receptor chimeras, DOR-D1(338–446), DOR-D1(368–446), and DOR-D1(379–400) were generated using PCR and homology-directed ligation (In-Fusion HD Cloning kit, Clontech). D1 oligos were fused to DOR residue 340. Flag-D1-PAGFP was generated using PCR and ligation into p-PAGFP-N1. Flag-SSTR3 was created using PCR and ligation into pIRES. Rab8a and Arl13b-YFP cDNAs were gifts from Jeremy Reiter (UCSF, San Francisco). Rab23, Rab23-S23N, Rab23-Q68L, and Rab11 cDNA were gifts from Keith Mostov (UCSF, San Francisco). Flag-Rab23-Q68L was created using PCR and ligation into pIRES. Receptor chimeras, Flag-D1Δ381-395-Rab23, Flag-D1Δ381-395-Rab23-S23N, Flag-D1Δ381-395-Rab23-Q68L, Flag-DOR-Rab23-Q68L, and Flag-DOR-Rab11-Q70L, Flag-DOR-Rab8-Q67L, Flag-MOR-Rab23-Q68L and Flag-B2AR-Rab23-Q68L, were generated using PCR and homology-directed ligation into pIRES (In-Fusion HD Cloning kit, Clontech). Rab23 constructs were fused to the C-terminal end of D1Δ381-395 with a 2 residue linker. Rab23-Q68L, Rab11-Q70L, and Rab8-Q67L were fused to the C-terminal end of DOR with an 8-residue linker.

Transfection of constructs was performed using Lipofectamine 2000 and RNAi-max (Invitrogen) for cDNA or siRNA, respectively, in accordance with manufacturer's instructions. Stably transfected cell clones expressing Flag-D1R were generated by selecting for neomycin resistance with 500 µg/ml G418 (Geneticin, Invitrogen). Target sequences for knockdown mIFT57 (1: 5′-CAGCAATTGGCTTC TATTAAA-3′, 2: 5′-TACAATGAATATAGTATTTAA-3′), mIFT172 (1: 5′-AAGGAGCATTTACAAGAA CAA-3′, 2: 5′-CCCACAGAATTTCAACATCTA-3′), mRab23 (1: 5′-AAGATTGGTGTCTTTAATGCA-3′, 2: 5′-TAGCCACTAAATGCATGGTAA-3′), and control (1027281, Qiagen). Duplex RNA (30 pm, Qiagen) was transfected into 40% confluent cells in a 6-well dish 72 hr before experimentation.

## Antibodies
Antibodies used were rabbit anti-Flag (Sigma), mouse anti-Flag M1 (Sigma), rat anti-HA (Roche Applied Science), and mouse anti-AcTub (Sigma).

## Co-immunoprecipitation
Cells expressing indicated constructs were grown to confluency in 10-cm dishes. 48 hr after transfection, cells were lysed in 0.2% Triton X-100, 200 mM NaCl, 50 mM Tris pH 7.4, and 1 mM $CaCl_2$ supplemented with a standard protease inhibitor mixture (Roche Applied Science) and cleared by centrifugation (12,000×$g$ for 10 min). Samples were pre-cleared by incubation with mouse IgG agarose (Sigma) at 4°C for 30 min. Samples were incubated with anti-Flag M2 affinity gel IgG (Sigma) at 4°C for 1 hr, washed with lysis buffer five times, and incubated with SDS sample buffer (Invitrogen) supplemented with dithiothreitol to elute proteins. Western immunoblot analysis was performed using rat anti-HA-peroxidase (Roche) or rabbit anti-Flag (Sigma). Immunoprecipitation signals were quantified by scanning densitometry of films exposed in the linear range. Linearity was verified by generating a standard curve using a dilution series of the indicated sample.

## Fixed cell microscopy
Cells were transfected with the indicated construct(s) and then plated on glass coverslips 24 hr later. Cells were grown to confluency to induce ciliation over 24 hr and then fixed. Surface Flag-tagged receptors were labeled by addition of rabbit anti-Flag antibody (1:500; Sigma) to the media for 20 min at 37°C. Cells were then washed with phosphate-buffered saline 2× and fixed by incubation in 4% formaldehyde (Fisher Scientific) diluted in PBS for 15 min at room temperature. Cells were permeabilized and blocked in 0.1% Triton X-100 and 3% milk diluted in PBS. Primary labeling of AcTub and HA was performed with mouse anti-AcTub (1:1000; Sigma) or rat anti-HA (1:1000; Roche Applied Science), respectively, for 1 hr. Secondary labeling was performed using donkey anti-rabbit Alexa Fluor 488 (1:1000; Invitrogen), goat anti-mouse Alexa Fluor 594 (1:1000; Invitrogen), and goat anti-rat Alexa Fluor 647 (1:1000; Invitrogen). Specimens were mounted using ProLong Gold antifade reagent (Life Technologies). Fixed cells were imaged by epifluorescence microscopy using a Nikon

inverted microscope, 60× NA 1.4 objective (Nikon), mercury arc lamp illumination, and standard dichroic filter sets (Chroma).

## Live cell microscopy

Cells were imaged at 37°C in Dulbecco's Modified Eagle Medium, D-MEM Glucose (DME H-21), w/o Phenol Red supplemented with 30 mM Hepes. Surface Flag-D1-PAGFP receptors were labeled by addition of mouse anti-Flag M1 antibody (1:500; Sigma) conjugated to Alexa Fluor 555. Cells were imaged on a spinning disk confocal microscope (Nikon TE-2000 with Yokogawa confocal scanner unit CSU22) using a 100× NA 1.45 objective. To photoactivate Flag-D1-PAGFP specifically in the cilium, 405-nm laser illumination was directed through a second light path via a single-mode fiber and focused in the image plane. Photoactivation was achieved by delivering brief (100 ms) pulses of 405-nm illumination.

## Microscope image acquisition

For epifluorescence microscopy of fixed cells, images were acquired using a cooled CCD camera (Princeton Instruments MicroMax) with illumination and exposure times adjusted to remain in the linear range of the camera. For spinning disc confocal microscopy of live cells, images were collected at 37°C using an electron multiplying CCD camera (Andor iXon 897) operated in the linear range. Images were processed at full bit depth for all analysis and rendered for display by converting to 8 bit format using ImageJ software (http://imagej.nih.gov/ij/) and a linear lookup table.

To measure the receptor fluorescence in the cilium, a region of interest (ROI) was manually created by outlining the cilium in the image. To correct for background fluorescence, the ROI was moved to a region outside the cell to measure representative fluorescence. This value was subtracted from the ciliary fluorescence. To measure the D1-PAGFP diffusion in cilium, the total PAGFP fluorescence was measured and normalized to the Flag-555 label to account for accumulation of receptor or focal plane (orientation) of the cilium.

To estimate lateral mobility of receptors in the cilium, the 405-nm laser spot was positioned so that it illuminated the center of the cilium but not the ends. A single photoactivation pulse was delivered with continuous confocal imaging at 0.5 Hz to monitor changes in the distribution of photoactivated Flag-D1-PAGFP in the cilium over time. To estimate new receptor delivery to the cilium, three 405-nm pulses were delivered over 10 s, determined empirically to photoactivate the majority of Flag-D1-PAGFP present in the cilium. We then acquired a subsequent GFP image, delivered another (single) 405-nm pulse, and acquired the GFP image again. This sequential 'image-photoactivate-image' sequence was applied either immediately (approximately 30 s) after the initial 405-nm pulse series or 30 min after. New delivery was estimated by the increment of integrated PA-GFP fluorescence intensity measured after the subsequent 405-nm pulse minus before, normalized to the integrated fluorescence intensity measured after. The anti-Flag Alex555 channel (unaffected by 405-nm pulses) was used to optimize focus on the cilium and to verify that the specimen did not move significantly between the 'before' and 'after' images. To assess the source of Flag-D1-PAGFP delivery to the cilium, we used a similar strategy as described above but quantified in the photoactivation series both PA-GFP and Alexa555 channels and determined their ratio. New receptor delivery from internal relative to plasma membrane sources was distinguished by changes in the PA-GFP/Alexa555 ratio, based on selective labeling with Alexa555 of only the plasma membrane pool, as discussed in text.

## Image analysis and statistical analysis

For line scan analysis, a straight line was drawn on the cilium, and the PlotProfile tool in ImageJ was used to determine the fluorescence intensity along the line. For integrated fluorescence determinations, the rectangular ROI tool was used. Results are displayed as mean of results from each experiment involving imaging of multiple specimens and cilia (specified in the figure legends). Error bars represent standard error of the mean based on at least n = 3 independent experiments done on different days unless noted otherwise. The statistical significance between conditions was analyzed using Student's unpaired $t$-test, calculated using Prism 6.0 software (GraphPad Software, Inc) and applying the Hochberg correction for multiple comparisons. The threshold for significance was $p < 0.05$ with exact p value ranges indicated in the figure legends.

## Fluorescence flow cytometry

Surface-accessible Flag immunoreactivity was quantified by fluorescence flow cytometry as described previously (*Tsao and Von Zastrow, 2000*). Briefly, cells were dissociated from culture dishes, labeled in suspension 4°C with anti-Flag M1 conjugated to Alexa 647, and analyzed using a FACS-Calibur instrument (Becton Dickenson). In each experiment, mean fluorescence intensity was determined from 10,000 cells and averaged over triplicate determinations for each construct. For all conditions shown, experiments were performed in triplicate, on separate days and from separate transfections. Error bars represent SEM across the experimental days. For determination of mutant receptor surface expression, transiently transfected cells were analyzed 48 hr after transfection. For evaluation of the effects of siRNA knockdown, stably transfected cells were analyzed 72 hr after transfection with the indicated siRNA duplex.

## Acknowledgements

We thank Aaron Marley for assistance at an early stage of this project and for generating the Flag-D1R stable cell line used in this study. We thank Dave Bryant, Wallace Marshall, Jeremy Reiter, and Kristen Verhey for valuable advice and generously providing reagents, and Henry Bourne, Roshanak Irannejad, Erik Jonsson, and Braden Lobingier for useful discussion. This work was supported by grants from the US National Institutes of Health (DA 010154, 010711 and 012864). AEL is a recipient of a predoctoral fellowship from the US National Science Foundation.

## Additional information

### Funding

| Funder | Grant reference | Author |
|---|---|---|
| National Institutes of Health (NIH) | DA010154 | Mark Von Zastrow |
| National Science Foundation (NSF) | predoctoral fellowship | Alison Leaf |
| National Institutes of Health (NIH) | DA010711 | Mark Von Zastrow |
| National Institutes of Health (NIH) | DA012864 | Mark Von Zastrow |

The funders had no role in study design, data collection and interpretation, or the decision to submit the work for publication.

### Author contributions

AL, Conception and design, Acquisition of data, Analysis and interpretation of data, Drafting or revising the article; MVZ, Conception and design, Analysis and interpretation of data, Drafting or revising the article

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
