## [Decision Letter]

Thank you for sending your work entitled “Dopamine receptors reveal an essential role of Rab23 as a ‘gatekeeper’ of cargo delivery to cilia” for consideration at *eLife*. Your article has been favorably evaluated by Randy Schekman (Senior editor) and three reviewers, one of whom is a member of our Board of Reviewing Editors.

The Reviewing editor and the other reviewers discussed their comments before we reached this decision, and the Reviewing editor has assembled the following comments to help you prepare a revised submission.

This work from the von Zastrow laboratory investigates targeting of the D1-type dopaminergic receptor (D1R) to the primary cilium of IMCD3 cells. Although protein trafficking to the cilium is well studied, this study indicates that new insights remain to be uncovered. The authors report that D1R has a unique cytoplasmic C-terminal sequence that appears to be required for targeting D1R; the C-terminal can induce ciliary targeting of the non-ciliary delta opiod peptide receptor (DOR). Using a small siRNA screen and dominant negative constructs, the authors report that D1R targeting also requires IFT-B, KIF17 and Rab23, and they show some data they interpret as a hierarchical regulation of these additional components. Overall, this is an interesting and well documented study that may be of broad interest. However, there are a number of criticisms that need addressing.

The reviewers were confused about what story the authors want to tell. On the one hand, the title of the manuscript indicates that the paper is about “Rab23 as a gatekeeper of ciliary entry” yet the data only implicate Rab23 in trafficking of D1R. On the other hand, the manuscript is about ciliary targeting of D1R and the roles of its targeting sequence and known ciliary transport factors such as IFT and KIF17.

With respect to the idea that Rab23 is a “gatekeeper of cargo delivery to cilia”, in the Discussion, the authors cite papers that Rab23 is not required for ciliary localization of polycystin-2 or Smo. Thus the idea that Rab23 is a “gatekeeper” doesn't seem to hold. The fact that Rab23 is not required for ciliary localization of Smo also indicates that Rab23 is not specialized for regulating lateral entry of membrane proteins but it would probably be good if the authors looked at Smo localization directly. Is it that Rab23 plays a role in ciliary localization of receptors with D1R-type targeting sequences? What about its role in ciliary localization of other receptors with known targeting sequences (e.g. SSTR3, MCHR1)?

Overall, the reviewers felt that the evidence for a ‘gate-keeper’ role for Rab23 was weak, and that the authors should concentrate on D1R ciliary targeting mechanisms.

A) Molecular mechanisms of D1R targeting.

1) Based on a C-terminal truncation analysis (Figure 3), the authors state that a short sequence (top page 9) is the ciliary targeting signal for D1R (aa 381-395). However, they then state that chimeras comprising DOR and the C-terminal of D1R caused DOR localization in the cilium, except when the C-terminal D1R sequence was truncated but still contained this putative DIR targeting sequence (Figure 3). It is unclear, therefore, how the authors can draw the conclusion that aa 381-395 is a ciliary targeting signal. Is the signal more complex, and involve a folded structure, part of which is regulated through by aa 381-395? Did the authors examine ciliary targeting of D1R that contained only aa 381-395 from the C-terminal cytoplasmic domain? Resolution of this problem has important relevance to other studies that used a deletion of this sequence, but not this sequence explicitly.

2) The authors should attempt to map the IFT-B-binding determinants on D1R and check whether they are congruent with the D1R CTS. Given that the authors have all the D1R truncations, the mapping of the interaction should be straightforward and will be informative. It would also be interesting to test interaction of IFT57 with known ciliary GPCRs.

3) With respect to the mechanisms by which D1R is targeted to the cilium, the reviewers liked the assays in Figures 1 and 2, but were concerned that assumptions regarding the photokinetics properties of of PAGFP could be misleading.

Figure 1 assumes that photoactivation of PA-GFP is complete and irreversible. While the photokinetics properties of PA-GFP in the test tube or other specific cellular environment do support the author's assumption, an independent verification that ciliary D1R-PAGFP is fully and irreversibly converted after 405 nm pulsing is essential. One possibility is that a fraction (e.g. 30%) of fluorophores enter a reversible dark state after the laser pulses. It should be straightforward to conduct the same experiment as in Figure 2 in fixed cells or after ATP depletion. This is an important point as reversible photoactivation and no transport within the 30 min time frame would account for the results in Figure 2.

The two predictions for a receptor that is delivered from a PM pool [1) a ratio PA/555 <1 at data point “30 min” and 2) a recovery to ratio PA/555 =1 at data point “30 min + PA”] could be obtained if the receptor was delivered from an internal pool and the PAGFP proportionally bleaches faster than the 555. The reviewers requested that more of the original data is shown with separate fluorescence channels. And some controls with no Flag antibody or unrelated 555 antibody.

In addition, a concern with the data presented in Figure 2 is the possibility of photo-damage to the transition zone by the repeated pulses at 405 nm. Photo-damage of the TZ may cause a rupture of the diffusion barrier thus allowing molecules to enter from the PM into the CM. This could cause the rather high recovery rate of D1R, compared to the GPCRs tested by others. There are assays for testing the functional integrity of the TZ by assaying exclusion of large (>200 kDa) proteins into cilia.

B) Roles of RAB23, Kif17 and IFT-B

1) Experiments fusing Rab23 to the D1R deficient in ciliary targeting. The reference about the G234A mutation (41) refers to a G234A mutant of conventional kinesin. Does mutation of G234A in KIF17 also result in a lack of ATP hydrolysis? Also, the work with the G234A mutant of conventional kinesin used recombinant homodimers of G234A whereas the G234A mutant KIF17 is over-expressed in cells that contain endogenous KIF17. How does this impact the ability of G234A KIF17 to work in cells? Do you get the same targeting defect upon knockdown of KIF17 or expression of a dominant negative motor?

2) The surface level of D1RΔ381-395-Rab23 is increased (Figure 7—figure supplement 1) while the cilia/PM ratio is not (Figure 7), indicating that the overall expression level of the protein is higher than the control (D1R). This should be discussed. To add weight to their model concerning the order of events (Figure 8), it would be good to show that GFP-Rab23-Q68L and/or Rab23 WT still localize to the cilium when Kif17-G234A is expressed. Also, can fusion of the WT or Q69L Rab23 protein to D1R rescue the trafficking effect upon knockdown of IFT or KIF17?

3) The Rab23-D1Rdelta[381-395] fusion is difficult to interpret: it is extremely artificial and suggests that combining two proteins that weakly localize to cilia can create a protein that strongly localize to cilia. Assigning a sequence of action of Rab23 based on this fusion seems like a stretch. To add weight to their model concerning the order of events (Figure 8), it would be good to show that GFP-Rab23-Q68L and/or Rab23 WT still localize to the cilium when Kif17-G234A is expressed or when KIF17 is depleted by siRNA.

4) The effect of KIF17 dominant negative are intriguing but somewhat difficult to interpret. For example, it is possible that over-expression of KIF17[G234A] interferes with IFT entry into cilia. Whitely authors could test the ciliary levels of various proteins (e.g. IFT88, ARL13B, BBS5, GPR161) in cells over-expressing KIF17[G234A], a better experiment would be to deplete Kif17 by siRNA.

Other points:

1) All knockdown studies should be performed with 2 different siRNAs.

2) Whole cell views need to be shown for every ciliary image and surface expression being quantified. And all expressed proteins need to be accounted for (e.g. is KIF17 or KIF17-G234A even expressed in the cells shown in Figure 5?).

3) All blots need to be quantified within the linear range. It seems possible that more IFT57 is co-precipitated with D1R in the Rab23 siRNA (Figure 8) but it is not possible to know without quantification across multiple experiments.

4) In general it is better to show large panels for fluorescence images and make the graphs small since these are easier to see.

5) The schematics in Figures 3, 6 and 7 would be easier to understand if the schematic included the whole protein and not just the C-terminal tail.

6) The inclusion of asterisks describing all significance levels in each of the figure legends is confusing as I was looking for asterisks in figures that did not contain them.

7) Figure 2: Curve fitting would represent the line scans better than scattered data points.

8) The scale bar in Figure 1 seems to have moved to Figure 1. There are no scales bars for any of the fluorescence images in Figures 2, 3, 4, 5, 6, 7 and 8. Some of the cilia look longer than others but it is impossible to know if this is real or an artifact of different image magnifications.

9) The reviewers found citations were missing:

i) Introduction, no mention of the BBSome or IFT-A in intro about targeting of GPCRs to cilia despite those papers being cited later in the Results section.

ii) Introduction, regarding lateral transport from the plasma membrane (PM) to the ciliary membrane (CM), the seminal paper from Hunnicutt, G. R., Kosfiszer, M. G., & Snell, W. J. (1990). The Journal of Cell Biology, 111(4), 1605-1616 should be cited.

iii) Introduction, the statement “If so, the mechanistic basis of this second route of ciliary receptor targeting remains a mystery” is exaggerated and fails to mention the hypothesis, and attendant citations, that the BBSome carries out lateral transport from plasma membrane to ciliary membrane.

---

## [Author Response]

*The reviewers were confused about what story the authors want to tell. On the one hand, the title of the manuscript indicates that the paper is about “Rab23 as a gatekeeper of ciliary entry” yet the data only implicate Rab23 in trafficking of D1R. On the other hand, the manuscript is about ciliary targeting of D1R and the roles of its targeting sequence and known ciliary transport factors such as IFT and KIF17*.

The reviewers are correct that, while we proposed Rab23 as a gatekeeper of membrane cargo entry to cilia, our studies focus on the D1R and include data regarding IFT-B and KIF17. Given new findings added to the present revision (see below), we do not think Rab23’s function is restricted to the D1R. However, we do not know how wide-ranging this requirement is for other membrane proteins. In considering the reviewers’ comments, we have changed the title in an effort to avoid implying a more general function than we can presently support. The new title is “Dopamine receptors reveal an essential role of IFT-B, KIF17, and Rab23 in delivering specific receptors to primary cilia.” We have taken the word ‘gatekeeper’ out of the title, but we still find it to be a useful term for explaining our results. However, we do not want to imply that Rab23 is a gatekeeper for all cargoes. Therefore, in the revised text (Abstract and Discussion), we have clarified the proposal that Rab23 is acting as a gatekeeper for select ciliary receptor cargoes.

With respect to the idea that Rab23 is a “gatekeeper of cargo delivery to cilia”, in the Discussion, the authors cite papers that Rab23 is not required for ciliary localization of polycystin-2 or Smo. Thus the idea that Rab23 is a “gatekeeper” doesn't seem to hold. The fact that Rab23 is not required for ciliary localization of Smo also indicates that Rab23 is not specialized for regulating lateral entry of membrane proteins but it would probably be good if the authors looked at Smo localization directly. Is it that Rab23 plays a role in ciliary localization of receptors with D1R-type targeting sequences? What about its role in ciliary localization of other receptors with known targeting sequences (e.g. SSTR3, MCHR1)?

Overall, the reviewers felt that the evidence for a ‘gate-keeper’ role for Rab23 was weak, and that the authors should concentrate on D1R ciliary targeting mechanisms.

With regard to the generality of Rab23 function, the reviewers suggested that we look at another cilia-localized receptor. We focused on SSTR3 because its ciliary delivery is well-documented, and because we used SSTR3 to assess the role of KIF17 function (Figure 5). Using the same strategy that is used to assess ciliary localization of D1R, we found that SSTR3 localization to cilia was similarly sensitive to Rab23 depletion. We have included these new data in Figure 6 and in the subsection “A discrete and essential function of Rab23 in the ciliary targeting mechanism”. Therefore, we are now confident that Rab23 function in cargo delivery to cilia is not restricted to the D1R, and we find this particularly interesting with regard to SSTR3, because ciliary localization of this GPCR is apparently KIF17-independent. While it is tempting to speculate about convergent mechanisms, we have not done so in the present study and intend to investigate more broadly the role of Rab23 in membrane cargo delivery to cilia in future studies. For the present study, as suggested by the reviewers, we have stepped back from making any statement on Rab23 generality and focus on D1R ciliary targeting mechanisms.

A) Molecular mechanisms of D1R targeting.

*1) Based on a C-terminal truncation analysis (*Figure 3*), the authors state that a short sequence (top page 9) is the ciliary targeting signal for D1R (aa 381-395). However, they then state that chimeras comprising DOR and the C-terminal of D1R caused DOR localization in the cilium, except when the C-terminal D1R sequence was truncated but still contained this putative DIR targeting sequence (*Figure 3*). It is unclear, therefore, how the authors can draw the conclusion that aa 381-395 is a ciliary targeting signal. Is the signal more complex, and involve a folded structure, part of which is regulated through by aa 381-395? Did the authors examine ciliary targeting of D1R that contained only aa 381-395 from the C-terminal cytoplasmic domain? Resolution of this problem has important relevance to other studies that used a deletion of this sequence, but not this sequence explicitly.*

The reviewers asked how we can draw the conclusion that aa 381-395 is a ciliary targeting determinant and that our data suggest that the ciliary targeting determinant is a more complex or extended structure. We wholeheartedly agree and did not mean to imply that aa 381-395 is the complete targeting determinant. Rather, our data indicate that aa 381-395 is an essential part of the determinant. We agree that the receptor chimera data suggests that the D1R ciliary targeting determinant is indeed more extensive, including residues in the proximal region of the D1R C-terminal tail. In response to the reviewer critiques, we have included specific evidence showing that fusion of a smaller sequence containing these essential residues (aa 379-400) is not sufficient by itself to drive ciliary localization of the non-ciliary DOR. These new data are included in Figure 3—figure supplement 5. We have noted this result in the subsection “Ciliary targeting of D1Rs is directed by an extended structural determinant located in the receptor’s cytoplasmic tail” and we have attempted to make this important distinction clearer in the revised manuscript. We changed the title of Figure 3 to “The D1R cytoplasmic tail is required for ciliary receptor targeting and does not resemble previously defined ciliary targeting determinants.”

2) The authors should attempt to map the IFT-B-binding determinants on D1R and check whether they are congruent with the D1R CTS. Given that the authors have all the D1R truncations, the mapping of the interaction should be straightforward and will be informative. It would also be interesting to test interaction of IFT57 with known ciliary GPCRs.

We agree that this is a reasonable hypothesis. To address this, we carried out additional experiments in which we examined the ability of the D1R C-terminal tail mutant, Flag-D1Δ381-395, which has reduced ciliary localization, to co-immunoprecipitate IFT57. This mutation did not prevent IFT57 co-IP and, to our surprise, it significantly increased the co-IP signal. Therefore, the functional activity of the D1R CTS is not directly congruent with IFT57 association. This is consistent with our multi-step model (accumulation of an intermediate), and we believe this provides evidence for a more complex operation of the D1R CTS. These new data, including quantification in the linear range, are included in Figure 4 and referred to in the revised text in the subsection “Ciliary targeting of D1Rs is promoted by IFT-B complex proteins and KIF17”.

We have also carried out additional experiments testing whether DOR fused to the entire D1R C-terminal tail (DOR-D1(338-446)) is able to physically associate with IFT57. We have observed a positive co-IP in some experiments but not consistently. We think this may indicate a weak interaction, but we are not comfortable making a definitive statement one way or the other in the present study.

*3) With respect to the mechanisms by which D1R is targeted to the cilium, the reviewers liked the assays in*
Figures 1 and 2*, but were concerned that assumptions regarding the photokinetics properties of of PAGFP could be misleading*.

Figure 1
*assumes that photoactivation of PA-GFP is complete and irreversible. While the photokinetics properties of PA-GFP in the test tube or other specific cellular environment do support the author's assumption, an independent verification that ciliary D1R-PAGFP is fully and irreversibly converted after 405 nm pulsing is essential. One possibility is that a fraction (e.g. 30%) of fluorophores enter a reversible dark state after the laser pulses. It should be straightforward to conduct the same experiment as in*
Figure 2
*in fixed cells or after ATP depletion. This is an important point as reversible photoactivation and no transport within the 30 min time frame would account for the results in*
Figure 2.

We appreciate this concern and agree that this is a conceivable problem. As the reviewer suggested, we have carried out additional controls in fixed cells. We do not see significant additional signal upon 405 nm illumination delivered 30 min after initial photoactivation. This contrasts with the behavior in live cells, where a significant increase is observed. This new data is included in Figure 2—figure supplement 4 and noted in the subsection “D1Rs are mobile in the ciliary membrane and accumulated by continuous delivery from the extra-ciliary plasma membrane pool”.

*The two predictions for a receptor that is delivered from a PM pool [1) a ratio PA/555 <1 at data point “30 min” and 2) a recovery to ratio PA/555 =1 at data point “30 min + PA”] could be obtained if the receptor was delivered from an internal pool and the PAGFP proportionally bleaches faster than the 555. The reviewers requested that more of the original data is shown with separate fluorescence channels. And some controls with no Flag antibody or unrelated 555 antibody*.

We agree that differential photobleaching is a potential problem with our quantitative imaging analysis. However, we do not think this is the case in the present experiments. In the revised manuscript, we have included data showing fluorescence intensities in both channels over a series of 28 consecutive acquisitions. This number of acquisitions is in excess of the number of acquisitions used in the experimental setup described in Figure 2 (typically 12 acquisitions in total). We do not see any significant photobleaching of either fluorophore under these conditions. These new data are included in Figure 2—figure supplement 6 and noted at the end of the subsection “D1Rs are mobile in the ciliary membrane and accumulated by continuous delivery from the extra-ciliary plasma membrane pool”.

As the reviewers suggested, we have included representative examples of primary image data for the experimental setup in Figure 2. These images are shown in Figure 2—figure supplement 5 and noted in the subsection “D1Rs are mobile in the ciliary membrane and accumulated by continuous delivery from the extra-ciliary plasma membrane pool”. We also include images showing specificity of the 555 signal and lack of bleed-through of the 488 channel into the 555 channel. This is shown in Figure 2—figure supplement 7 and described in the text at the end of the aforementioned subsection.

*In addition, a concern with the data presented in*
Figure 2
*is the possibility of photo-damage to the transition zone by the repeated pulses at 405 nm. Photo-damage of the TZ may cause a rupture of the diffusion barrier thus allowing molecules to enter from the PM into the CM. This could cause the rather high recovery rate of D1R, compared to the GPCRs tested by others. There are assays for testing the functional integrity of the TZ by assaying exclusion of large (>200 kDa) proteins into cilia*.

The reviewers raised the concern that our photoactivation strategy could damage the transition zone, allowing molecules to enter from the PM into the ciliary membrane. The concern is that this could cause an inappropriately high delivery rate.

We agree, in principle, that rupture of the diffusion barrier could confound our experiments. However, in practice, PA-GFP photoactivation requires a much lower light dose than GFP FRAP. Further, if the diffusion barrier was disrupted, we would expect D1Rs concentrated in the cilium to diffuse out, equilibrating by mass action with the extra-ciliary plasma membrane pool. This is clearly not the case. As part of the new data added to Figure 2—figure supplement 5**,** one can see ciliary D1R concentration is maintained for 30 min after the photoactivation and multiple image acquisitions. Additionally, we now include new data (Figure 2—figure supplement 3) showing that the PA-GFP fluorescence increment increases with time after the initial photoactivation series (noted in the subsection “D1Rs are mobile in the ciliary membrane and accumulated by continuous delivery from the extra-ciliary plasma membrane pool”). This further argues against rupture of the diffusion barrier and supports the conclusion that PA-GFP fluorescence increment is due to time-dependent delivery of new (i.e. from outside the cilium) D1Rs.

*B) Roles of RAB23*, *Kif17 and IFT-B*

*1) Experiments fusing Rab23 to the D1R deficient in ciliary targeting. The reference about the G234A mutation (*[41]*) refers to a G234A mutant of conventional kinesin. Does mutation of G234A in KIF17 also result in a lack of ATP hydrolysis? Also, the work with the G234A mutant of conventional kinesin used recombinant homodimers of G234A whereas the G234A mutant KIF17 is over-expressed in cells that contain endogenous KIF17. How does this impact the ability of G234A KIF17 to work in cells? Do you get the same targeting defect upon knockdown of KIF17 or expression of a dominant negative motor?*

In considering the reviewer’s question, we recognize that we failed to specify the nature of the motor domain mutation. The mutation is in switch II, the “relay” domain that is essential for kinesin movement. While kinesin motor domains are quite variable in other regions, switch II is highly conserved. In fact, switch II in KIF17 and kinesin-1 are identical, and G234 corresponds to precisely the critical residue for motor movement that was defined previously through study of kinesin-1. We clarify in the text of the revised manuscript in subsection “Ciliary targeting of D1Rs is promoted by IFT-B complex proteins and KIF17” and include sequence alignment in Figure 5—figure supplement 1.

The reviewers make a good suggestion to examine the effect of the KIF17 knockdown. We attempted to do so and despite trying 4 independent siRNAs, we were unable to obtain successful knockdown of KIF17 as assessed by qPCR. Therefore, as an alternative approach to obtain independent evidence for KIF17’s role in D1R ciliary localization, we tested the effect of a different and previously described dominant negative KIF17 construct (27). We include this new data in the revised manuscript (Figure 5—figure supplement 3) showing that expression of KIF17-DN decreases D1R ciliary enrichment. These new data are explicitly noted at the end of the aforementioned subsection.

*2) The surface level of D1RΔ381-395-Rab23 is increased (*Figure 7—figure supplement 1*) while the cilia/PM ratio is not (*Figure 7*), indicating that the overall expression level of the protein is higher than the control (D1R). This should be discussed. To add weight to their model concerning the order of events (*Figure 8*), it would be good to show that GFP-Rab23-Q68L and/or Rab23 WT still localize to the cilium when Kif17-G234A is expressed*. *Also, can fusion of the WT or Q69L Rab23 protein to D1R rescue the trafficking effect upon knockdown of IFT or KIF17?*

Surface expression:

The reviewer is correct but we do not think that this is a significant confound because the reverse is true with DOR fused to Rab23 (Figure 7—figure supplement 4). Therefore the ciliary localization effect of Rab23 is independent of its effect on overall surface expression. We have explicitly noted and discussed this point in the revised text (in subsection “A discrete and essential function of Rab23 in the ciliary targeting mechanism”).

Rab23 QL localization:

We have added new data testing whether or not KIF17-G234A affects Rab23-Q68L ciliary localization. We found that it does not, further corroborating our interpretation. This new data is shown in Figure 8—figure supplement 2 (quantification in the figure legend) and discussed in the subsection “A discrete and essential function of Rab23 in the ciliary targeting mechanism”**.**

Rab23-Q68L fusion rescue:

Additionally we tested whether fusion of the Rab23-Q68L to D1R (Flag-D1Δ381-395-Rab23-Q68L) can rescue the trafficking defect upon knockdown of IFT172. Indeed, the Rab23-Q68L fusion effectively bypasses the requirement for IFT172 as shown new data added in Figure 8 of the revised manuscript and discussed in the subsection “A discrete and essential function of Rab23 in the ciliary targeting mechanism”**.**

*3) The Rab23-D1Rdelta[381-395] fusion is difficult to interpret: it is extremely artificial and suggests that combining two proteins that weakly localize to cilia can create a protein that strongly localize to cilia. Assigning a sequence of action of Rab23 based on this fusion seems like a stretch. To add weight to their model concerning the order of events (*Figure 8*), it would be good to show that GFP-Rab23-Q68L and/or Rab23 WT still localize to the cilium when Kif17-G234A is expressed or when KIF17 is depleted by siRNA*.

We agree that these fusion constructs are artificial. However, the effects are strong and unambiguous. We showed in the original submission that Rab23QL fusion is also sufficient to confer ciliary localization on a distinct receptor, DOR, which is normally excluded from cilia altogether. In the revised manuscript we add new data showing the same effect for two other non-ciliary GPCRs, MOR and B2AR (Figure 7—figure supplement 5, explained in the subsection “A discrete and essential function of Rab23 in the ciliary targeting mechanism”). Furthermore, we add new data establishing that this targeting effect is specific to Rab23. We already showed that Rab11 fusion does not confer ciliary localization. In the revised manuscript we add new data showing that fusion of Rab8 is also not sufficient to drive ciliary receptor localization (Figure 7 and explained in the aforementioned subsection). We think this highly significant because Rab8 by itself can clearly localize to cilia. Finally, we included the control experiment looking at Rab23-Q68L ciliary localization in the presence of KIF17-G234A (see above). We think that these four pieces of new data, taken together, add weight to our proposed model and, in particular, strongly support a specific ciliary targeting function of Rab23.

*4) The effect of KIF17 dominant negative are intriguing but somewhat difficult to interpret. For example, it is possible that over-expression of KIF17[G234A] interferes with IFT entry into cilia. Whitely authors could test the ciliary levels of various proteins (e.g. IFT88, ARL13B, BBS5, GPR161) in cells over-expressing KIF17[G234A], a better experiment would be to deplete Kif17 by siRNA*.

We agree that a KIF17 dominant negative could, in principle, disrupt ciliary D1R targeting by blocking overall cargo entry, and that KIF17 knockdown would be useful. As noted above, we found KIF17 mRNA resistant to numerous duplexes. However, we are confident that the mutant KIF17 is not producing an overall entry defect because ciliary targeting of SSTR3 is unaffected by mutant KIF17. This is shown in Figure 5. It remains possible that one or more of the other ciliary proteins that the reviewer notes depend on KIF17, but the present SSTR3 results, we believe, rule out an overall cargo entry defect.

*Other points*:

*1) All knockdown studies should be performed with 2 different siRNAs*.

We have done so by adding an additional siRNA for Rab23 and an additional siRNA for IFT57. These data are included in Figure 4 and Figure 6 and Figure 6 in the revised manuscript.

*2) Whole cell views need to be shown for every ciliary image and surface expression being quantified. And all expressed proteins need to be accounted for (e.g. is KIF17 or KIF17-G234A even expressed in the cells shown in*
Figure 5*?)*.

We have done so. In the revised manuscript we have included whole cell image views as supplemental figures for every cilium shown. Additionally we show expression of KIF17 and KIF17-G234A in Figure 5—figure supplement 3, Figure 5—figure supplement 5, and Figure 8—figure supplement 1. We show expression of IFT57 in Figure 4—figure supplement 1.

*3) All blots need to be quantified within the linear range. It seems possible that more IFT57 is co-precipitated with D1R in the Rab23 siRNA (*Figure 8*) but it is not possible to know without quantification across multiple experiments*.

We have quantified blots in the linear range, as the reviewer indicated, and from multiple experiments (n = 3 for each). Rab23 knockdown indeed increased the IFT57 co-IP signal (shown in Figure 8), as the reviewer noticed from the single example shown. However, this effect was variable and did not reach significance in our data set. The new data added to investigate congruence between IFT57 co-IP and ciliary localization, however, did reveal a significant effect. Deletion of residues 381-395 in the D1R tail, although this reduced ciliary receptor localization, significantly increased IFT57 co-IP (shown in Figure 4). We thank the reviewer for suggesting this experiment that, as also discussed above, adds weight to our interpretation and supports the multi-step model proposed in Figure 8.

*4) In general it is better to show large panels for fluorescence images and make the graphs small since these are easier to see*.

We have addressed this. Please see comment 2 above.

*5) The schematics in*
Figures 3, 6 and 7
*would be easier to understand if the schematic included the whole protein and not just the C-terminal tail*.

In the revised manuscripts, all the schematics have been modified as the reviewer suggests.

*6) The inclusion of asterisks describing all significance levels in each of the figure legends is confusing as I was looking for asterisks in figures that did not contain them*.

We have corrected this.

*7)*
Figure 2*: Curve fitting would represent the line scans better than scattered data points*.

We have added lines connecting the points.

*8) The scale bar in*
Figure 1
*seems to have moved to*
Figure 1*. There are no scales bars for any of the fluorescence images in*
Figures 2, 3, 4, 5, 6, 7 and 8*. Some of the cilia look longer than others but it is impossible to know if this is real or an artifact of different image magnifications*.

This is complete. We thank the reviewers for noticing this and in the revised manuscript we verified that all figures have scale bars.

*9) The reviewers found citations were missing*:

*i) Introduction, no mention of the BBSome or IFT-A in intro about targeting of GPCRs to cilia despite those papers being cited later in the Results section*.

*ii) Introduction, regarding lateral transport from the plasma membrane (PM) to the ciliary membrane (CM), the seminal paper from Hunnicutt, G. R., Kosfiszer, M. G., & Snell, W. J. (1990). The Journal of Cell Biology, 111(4), 1605-1616 should be cited*.

*iii) Introduction, the statement “If so, the mechanistic basis of this second route of ciliary receptor targeting remains a mystery” is exaggerated and fails to mention the hypothesis, and attendant citations, that the BBSome carries out lateral transport from plasma membrane to ciliary membrane*.

We thank the reviewers for noticing these missing citations and we have incorporated them into the paper accordingly.